# Electro-assisted methane oxidation to formic acid via in-situ cathodically generated $H_2O_2$ under ambient conditions

Jiwon Kim[1,2,7], Jae Hyung Kim[3,7], Cheoulwoo Oh[1,2], Hyewon Yun[4,5], Eunchong Lee[4], Hyung-Suk Oh [2,6], Jong Hyeok Park [1] ✉ & Yun Jeong Hwang [4,5] ✉

Direct partial oxidation of methane to liquid oxygenates has been regarded as a potential route to valorize methane. However, $CH_4$ activation usually requires a high temperature and pressure, which lowers the feasibility of the reaction. Here, we propose an electro-assisted approach for the partial oxidation of methane, using in-situ cathodically generated reactive oxygen species, at ambient temperature and pressure. Upon using acid-treated carbon as the electrocatalyst, the electro-assisted system enables the partial oxidation of methane in an acidic electrolyte to produce oxygenated liquid products. We also demonstrate a high production rate of oxygenates (18.9 μmol h$^{-1}$) with selective HCOOH production. Mechanistic analysis reveals that reactive oxygen species such as •OH and •OOH radicals are produced and activate $CH_4$ and $CH_3OH$. In addition, unstable $CH_3OOH$ generated from methane partial oxidation can be additionally reduced to $CH_3OH$ on the cathode, and so-produced $CH_3OH$ is further oxidized to HCOOH, allowing selective methane partial oxidation.

Methane, the main component of natural gas, has been an important resource for energy and chemical production owing to its abundance and high energy density (55 MJ kg$^{-1}$)[1,2]. Unfortunately, the utilization of natural gas has been hampered by the difficulty of its long-distance transportation in gaseous form, and unavailable methane has been flared into $CO_2$, a form of less potent greenhouse gas, in oil fields (143 billion m$^3$ of natural gas was flared in 2020 globally)[3,4]. The methane partial oxidation to liquid oxygenates such as $CH_3OH$ and HCOOH has been regarded as a promising approach for effective natural gas utilization as it can facilitate transportation and storage and mitigate $CO_2$ emission[5–7]. However, methane is a stable non-polar hydrocarbon ($\Delta H_{C-H} = 439.3$ kJ mol$^{-1}$) that is challenging to activate[8]. Moreover, the

complete oxidation of methane to $CO_2$ is more favorable than its partial oxidation (Eqs. (1, 2)). Hence, it is still challenging to induce the partial oxidation of methane without over-oxidizing it to $CO_2$.

$$CH_4(g) + \tfrac{1}{2}O_2(g) \rightarrow CH_3OH(l) \quad \triangle G^o = -114 \text{ kJmol}^{-1} \quad (1)$$

$$CH_4(g) + 2O_2(g) \rightarrow CO_2(g) + 2H_2O(l) \quad \triangle G^o = -804 \text{ kJmol}^{-1} \quad (2)$$

For the production of methanol or other liquid products from methane, the current industrial process relies on the following two-step indirect route: steam methane reforming (SMR) to syngas

[1]Department of Chemical and Biomolecular Engineering, Yonsei-KIST Convergence Research Institute, Yonsei University, 50 Yonsei-ro, Seodaemun-gu, Seoul 120-749, Republic of Korea. [2]Clean Energy Research Center, Korea Institute of Science and Technology, Hwarang-ro 14-gil 5, Seoul 02792, Republic of Korea. [3]Clean Fuel Research Laboratory, Korea Institute of Energy Research, Daejeon 34129, Republic of Korea. [4]Department of Chemistry, Seoul National University, Seoul 08826, Republic of Korea. [5]Center for Nanoparticle Research, Institute for Basic Science (IBS), Seoul 08826, Republic of Korea. [6]KIST-SKKU Carbon-Neutral Research Center, Sungkyunkwan University (SKKU), Suwon 16419, Republic of Korea. [7]These authors contributed equally: Jiwon Kim, Jae Hyung Kim. ✉e-mail: lutts@yonsei.ac.kr; yjhwang1@snu.ac.kr

(CO and $H_2$) at a high temperature (700–1000 °C) and pressure (5–40 bar) followed by a gas-to-liquid process[9]. Since SMR process has high emission of $CO_2$, at almost 9 kg of $CO_2$ per 1 kg of $H_2$ produced, and requires energy-intensive and large infrastructure, it is infeasible at remote geographical locations[10,11]. Therefore, energy-efficient, low $CO_2$ emission, and miniaturized technologies are highly required for direct partial oxidation of methane.

To date, many approaches have been developed for direct methane partial oxidation via thermocatalytic reactions; however, they need to be improved in terms of both efficiency and catalyst/oxidant candidates. Early studies employed liquid-phase homogeneous catalysis using complexes of low-valence precious metals (Hg, Pt, and Au) with a strong acid, such as oleum[12–14]. These systems, however, require highly corrosive solvents and additional hydrolysis steps for product extraction. In later studies, bioinspired Cu- or Fe-exchanged zeolite catalysts were exploited with $N_2O$, $O_2$, or $H_2O$ oxidants for stepwise methane conversion in the gas phase[15–17]. Although high selectivity was achieved, such conversions suffer from low reaction rates, high reaction temperature (300–550 °C), and catalyst regeneration step. Recently, an aqueous-phase partial oxidation of methane over Fe and Rh catalysts using $H_2O_2$ as the oxidant was reported[18–20]. Although higher production yield and selectivity were achieved, the high cost of $H_2O_2$, which is more expensive than the reaction products, limits the practical application of this process. Therefore, efforts have been made to convert methane to oxygenates using in-situ $H_2O_2$[5,21,22] generated from $O_2$ and $H_2$ over a catalyst such as AuPd or PdCu[5,21]. However, such conversions require $H_2$ as a co-reductant, precious metal catalysts, and elevated temperatures (70–120 °C). Hutchings et al. reported an Au-ZSM-5 catalyst using $O_2$ as the sole oxidant for the partial oxidation of methane; however, the reaction conditions were still harsh (120–240 °C and 23.2 bar)[22]. For conducting partial oxidation of methane in industrial areas, systems with mild reaction conditions should be developed using earth-abundant materials and eco-friendly oxidants.

While thermocatalytic oxidation reactions often require a high temperature and pressure, electrochemical oxidation can be conducted under ambient conditions. It offers a sustainable route for converting $CH_4$ to liquid fuels, mitigating $CO_2$ emissions[23–29]. For electrochemical methane oxidation, the formation of active oxygen species (e.g., $O^{2-}$ and $O^-$) on the electrocatalysts is necessary. Bertazzoli et al. succeeded in producing $CH_3OH$, $HCHO$, and $HCOOH$ as liquid products from methane using $TiO_2/RuO_2$ and $V_2O_5/TiO_2/RuO_2$ electrocatalysts[23,24]. NiO/Ni catalyst has also been reported to generate $C_2H_5OH$, and $CH_3OH$ in an H-cell[26]. However, the production rates of oxygenates were insufficient possibly due to the over-oxidation of the products to $CO_2$ or the competitive oxygen evolution reaction. To resolve these issues at the anode, electro-assisted methane partial oxidation (EMPO) at the cathode side has been proposed[27–29]. By reacting methane with the active oxygen species generated from the electrochemical oxygen reduction reaction (ORR), methane partial oxidation could be accomplished. However, there are only few reports about EMPO systems, and the reaction mechanism of EMPO is unclear.

In this study, we systematically investigate the EMPO system and demonstrate its promising production rate of oxygenates for methane partial oxidation. The EMPO system involves both electrochemical and chemical reactions. In this system, $H_2O_2$ is electrochemically produced on the cathode through a two-electron pathway ORR, and the $H_2O_2$ production involves oxidizing methane in the catholyte. The EMPO process can provide a high oxygenates production rate of 18.9 μmol $h^{-1}$ at room temperature and atmospheric pressure. Mechanistic investigations indicate that the EMPO system is beneficial for selective methane oxidation to HCOOH. Reactive oxygen species (ROSs) (i.e., •OH and •OOH radicals) are generated during ORR and activated $CH_4$ and $CH_3OH$. $CH_3OOH$ is one of the major products in methane partial oxidation; however, it is not a suitable liquid fuel for long-distance

transportation because of its instability. In the EMPO system, most of the produced $CH_3OOH$ is electrochemically reduced to $CH_3OH$ on the cathode, and $CH_3OH$ is further oxidized to HCOOH, resulting in high selectivity to HCOOH (80.7%). We also demonstrate that the electro-assisted oxidation process can be a general approach for the partial oxidation of small hydrocarbons such as ethane. We envisage that this miniaturized and sustainable system can provide a beneficial strategy for direct methane partial oxidation.

## Results

### Feasibility study of EMPO system

First, an acid-treated ketjen black (a-KB) carbon powder catalyst was prepared as a selective cathode catalyst for generating $H_2O_2$ via the two-electron pathway ORR[30,31]. Pristine ketjen black (KB) powder was acid-treated at an elevated temperature (80 °C) to introduce oxygen functional groups (−COOH and C−O−C), which are active in two-electron ORR. X-ray photoelectron spectroscopy (XPS) was performed to examine the changes in oxygenated species on the surface of the acid-treated carbon powder. (Supplementary Fig. 1). The C 1s spectrum of a-KB could be deconvoluted into bands attributed to the following species: graphitic carbon (C−C) at 284.8 eV, defects attributed to amorphous carbon at 285.9 eV, carbon singly bound to oxygen (C−O) at 286.8 eV, carbon bound to two oxygen atoms (i.e., −COOH) at 288.8 eV, and carbon in an aromatic ring at 291.2 eV (π−π* transitions)[32,33]. Deconvolution of the O 1s peak resulted in two peaks: oxygen doubly bound to carbon (C=O) at 533.7 eV and oxygen singly bound to carbon (C−O) at 532.1 eV[34]. Both the C 1s and O 1s signals indicate an increase in oxygenated species with C−O and C=O functional groups (e.g., C−OH, C−O−C, and O=C−OH) after acid treatment. High-resolution transmission electron microscopy (HR-TEM) and energy-dispersive X-ray spectroscopy (EDS) mapping images of a-KB revealed that the oxygen functional groups were well distributed over the surface of carbon (Supplementary Fig. 2). The ORR performance of a-KB was then evaluated in an aqueous solution (0.05 M $H_2SO_4$) using the rotating ring-disk electrode (RRDE) technique. The polarization curves of the catalyst (Supplementary Fig. 3) indicate that the onset potential is -0.3 V (vs. reversible hydrogen electrode (RHE)). Furthermore, the a-KB catalyst exhibited a high $H_2O_2$ selectivity of >90% over the applied potential ranges (0–0.3 V vs. RHE).

Next, to demonstrate the proposed EMPO system mediated by in-situ cathodic $H_2O_2$ generation, we conducted the reaction in an H-type cell (Fig. 1a). The working electrode was prepared by spraying a-KB based catalyst ink on carbon paper. The EMPO was carried out in a 0.05 M $H_2SO_4$ electrolyte with continuous feeding of $O_2$ and $CH_4$ gases. The potential was maintained at 0 V (vs. RHE), at which $H_2O_2$ was selectively produced from ORR (Supplementary Fig. 3), and the reaction was conducted at 25 °C for 30 min. $^1H$ and $^{13}C$ nuclear magnetic resonance (NMR) spectra were obtained to analyze the liquid products using an internal standard. Figure 1b shows that $CH_3OH$, $CH_3OOH$, and HCOOH were generated as liquid products of the partial oxidation of methane. No liquid products were detected when Ar was fed instead of $CH_4$[18,35]. The isotope experiment using $^{13}CH_4$ corroborated that $H^{13}COOH$ originated from the methane partial oxidation, as shown in Fig. 1c and Supplementary Fig. 4 ($CH_3OH$ and $CH_3OOH$ could not be detected because of the sampling method)[36]. We conducted an EMPO reaction applying Au foil as a carbon-free ORR catalyst, and the HCOOH was also generated as a liquid product confirming that the reaction products are generated from $CH_4$ oxidation (Supplementary Fig. 5). Then we analyzed time-dependent product formation under 0 V (vs. RHE), which further verified that the amount of the major product, HCOOH, increased linearly with the reaction time (Fig. 1d), supporting the continuous EMPO process. The amounts of minor products, $CH_3OOH$ and $CH_3OH$, increased to a certain level and remained constant. The different behaviors of time-dependent products' amounts originate from the reaction mechanism of EMPO,

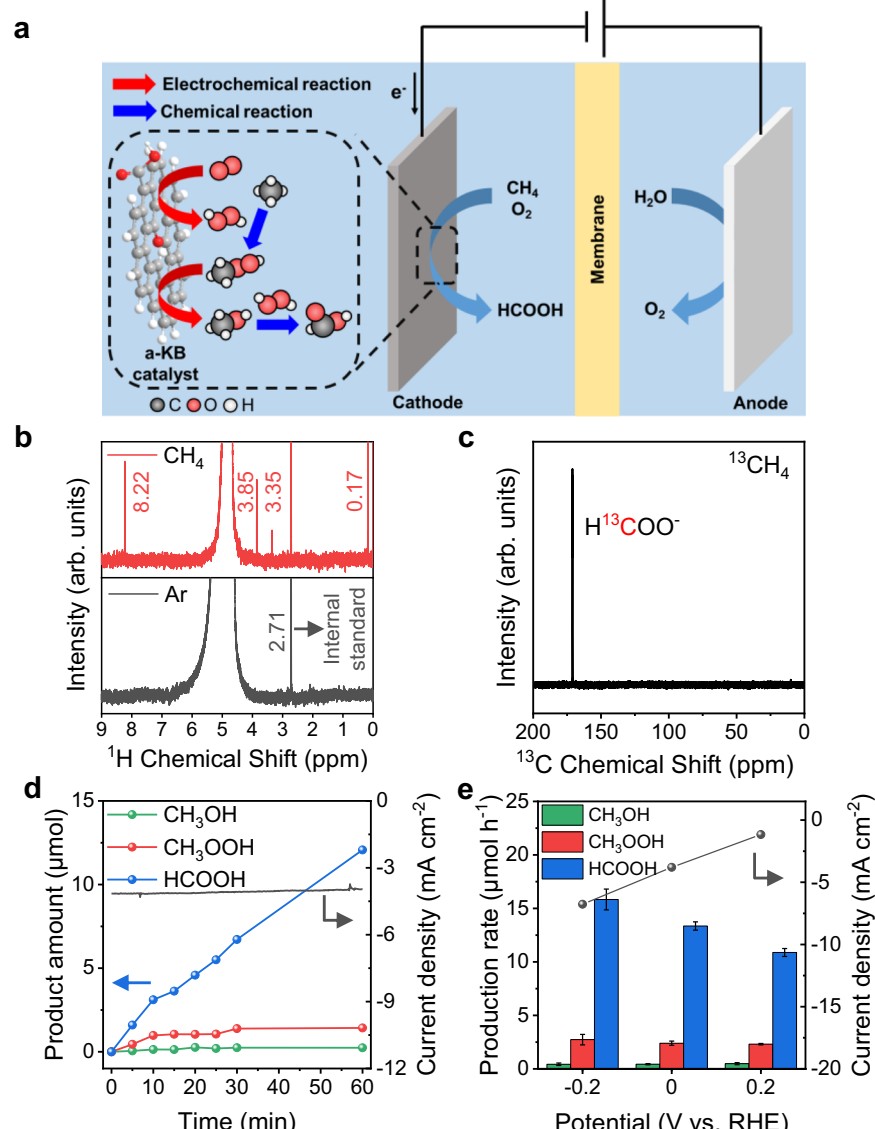

**Fig. 1 | Electro-assisted selective CH$_4$ partial oxidation system. a** Schematic illustration of the EMPO system. **b** $^1$H-NMR spectra of the products after reaction with purging CH$_4$ (red) or Ar (gray) along with O$_2$. The peaks at 0.17, 2.71, 3.35, 3.85, and 8.22 ppm are attributed to CH$_4$, (CH$_3$)$_2$SO (DMSO), CH$_3$OH, CH$_3$OOH, and HCOOH, respectively. **c** $^{13}$C-NMR spectrum of the product after EMPO using $^{13}$CH$_4$.

The peak at 166 ppm is attributed to H$^{13}$COO$^-$. **d** Time-dependent change in the amounts of liquid products of EMPO. **e** Potential-dependent production rates of liquid products of EMPO. Reaction conditions: 25 °C, 1 bar, 30 min, O$_2$: 100 sccm, CH$_4$: 100 sccm, 55 mL of 0.05 M H$_2$SO$_4$, and stirring at 700 rpm. The error bars represent the standard deviation. Source data are provided as a Source Data file.

which is later discussed. Additionally, we investigated whether gas products such as CO and CO$_2$ were generated due to the over-oxidation of CH$_4$ by on-line gas chromatography (GC). During the EMPO reaction, the amount of the produced CO$_2$ was 0.76 μmol h$^{-1}$ which was an insignificant amount compared to that of HCOOH (13.8 μmol h$^{-1}$) (Supplementary Figs. 6, 7).

Then other parameters, which can affect production rates, such as flow rate and applied potential were examined. First, we increased the flow rates of both feeding gases (CH$_4$ and O$_2$) from 100 sccm to 200 sccm (Supplementary Fig. 8a). The same product amounts were observed meaning that these total flow rates do not have an impact on the EMPO reaction rate (i.e., sufficient supply of reactant gases). On the other hand, the production of HCOOH is influenced by changes in the partial concentrations of CH$_4$ and O$_2$ (Supplementary Fig. 8b). To control the CH$_4$:O$_2$ ratio, we reduced the CH$_4$ flow rate to 25 sccm while maintaining the O$_2$ flow rate (resulting in a decrease in the partial pressure of CH$_4$), which led to a decrease in the amount of HCOOH produced. In contrast, when the flow rate of O$_2$ was reduced to 25 sccm

while keeping the CH$_4$ flow rate constant, the production of HCOOH increased. As the flow rate of CH$_4$/O$_2$ changed from 25 sccm/100 sccm to 50 sccm/50 sccm to 100 sccm/25 sccm, the partial concentration of CH$_4$ increased from 20% to 50% to 80%. The amount of HCOOH produced increased in the order of increasing partial concentration of CH$_4$, indicating that CH$_4$ was the limiting reagent and affected the reaction rate under these EMPO conditions.

In addition, the production rate depended on the applied potential (Fig. 1e) and is related to the amount of cathodically generated H$_2$O$_2$. As the applied potential was swept from +0.2 to −0.2 V (vs. RHE) (i.e., as the cathodic current increased), the amounts of oxygenated liquid products increased due to increased H$_2$O$_2$ generation. The highest production rate of the total oxygenates was 18.9 μmol h$^{-1}$ (CH$_3$OH: 0.4 μmol h$^{-1}$, CH$_3$OOH: 2.7 μmol h$^{-1}$, and HCOOH: 15.8 μmol h$^{-1}$). Although the production rate increased at −0.2 V (vs. RHE), the hydrogen evolution reaction can occur competitively below 0 V (vs. RHE) in an acidic electrolyte. Therefore, we performed the subsequent experiments at 0 V (vs. RHE).

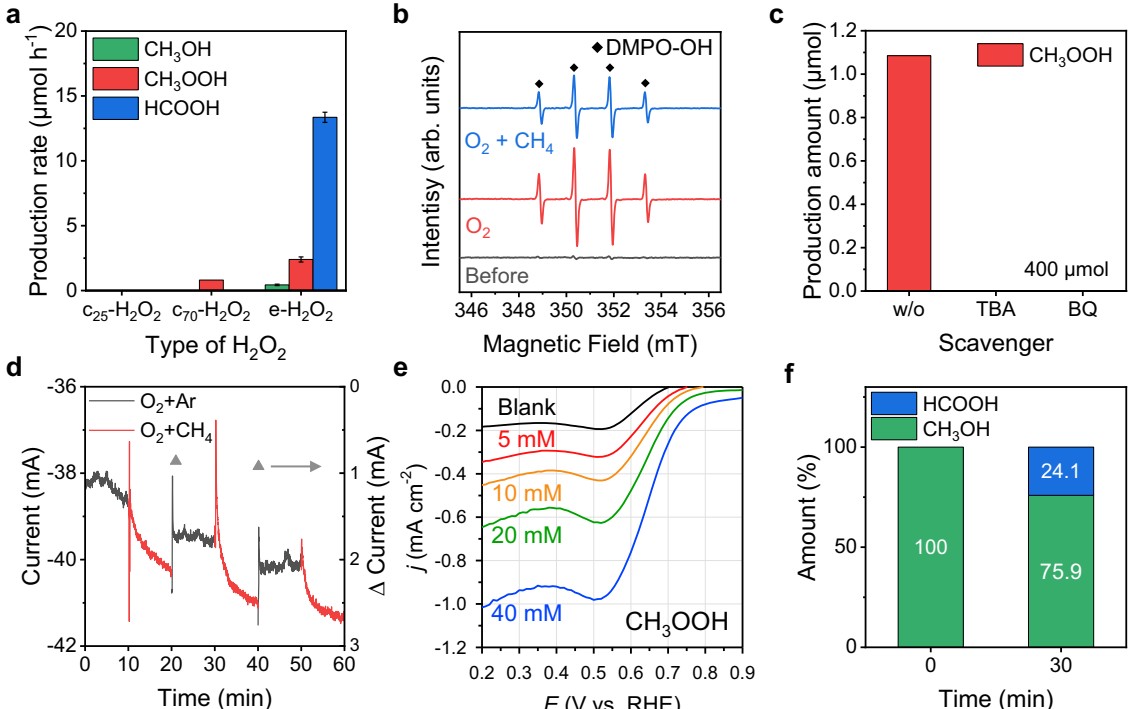

**Fig. 2 | Electrochemical analysis of EMPO. a** Performance comparison of a system using commercial $H_2O_2$ (c$_{25}$-$H_2O_2$ and c$_{70}$-$H_2O_2$ indicate reactions conducted at 25 °C and 70 °C, respectively) and the EMPO system (e-$H_2O_2$). The error bars represent the standard deviation. **b** EPR spectra of the electrolytes before (gray line) and after applying potential (0 V vs. RHE) under $O_2(g)$ (red line) and $O_2(g)$+$CH_4(g)$ (blue line) flowing with DMPO. **c** $CH_3OOH$ production after EMPO reaction in the presence of 400 μmol of scavengers. TBA and BQ are radical scavengers that can trap •OH and •OOH, respectively. **d** Chronoamperometry under $O_2$ + Ar and $O_2$ + $CH_4$ purging at 0 V (vs. RHE); $O_2$, Ar, and $CH_4$: 100 sccm. **e** Polarization curves in the presence of 0–40 mM of $CH_3OOH$ in a 0.05 M $H_2SO_4$ electrolyte. The black, red, orange, green, and blue lines represent blank, 5, 10, 20, and 40 mM conditions, respectively. **f** Result of methanol oxidation using ORR. Source data are provided as a Source Data file.

## Investigation of the reaction mechanism of EMPO

Next, we explored the reaction mechanism of EMPO mediated by in-situ generated $H_2O_2$. First, we compared the performances of the EMPO system (e-$H_2O_2$) and a non-electro-assisted system using commercial $H_2O_2$ (c-$H_2O_2$) as the oxidant (Fig. 2a). In the c-$H_2O_2$ system, c-$H_2O_2$ was supplied with a syringe pump, and the injection rate was controlled to be the same as the $H_2O_2$ production rate estimated from the 2e$^-$ ORR activity in our EMPO system (calculated based on the RRDE analysis). Other reaction conditions such as the electrolyte (0.05 M $H_2SO_4$) and reaction time (30 min) were the same, except for the feeding of $O_2$ gas. In contrast with EMPO, no liquid products were formed when c-$H_2O_2$ was added to the reaction cell at 25 °C (Fig. 2a and Supplementary Figs. 9, 10) implying that $H_2O_2$ molecule itself cannot react with $CH_4$ and ROSs formed during electrochemical ORR are crucial for the EMPO[37,38].

We performed control experiments to figure out whether ROSs are generated from $O_2$ and participate in the EMPO reaction. To understand the role of $O_2$ in the EMPO reaction, we applied the same cathodic potential in the H-cell under $CH_4$ and Ar flow. No $CH_4$ oxidation products were detected after the reaction feeding $CH_4$ + Ar except $O_2$ (Supplementary Fig. 11), supporting that O atoms from $O_2(g)$ not from the $H_2O$ electrolyte participate in activating $CH_4$. Then, to verify radical species more directly, we additionally conducted electron paramagnetic resonance (EPR) analysis by adding 5,5-dimethyl-1-pyrroline N-oxide (DMPO) in the electrolytes as a spin trap. EPR signals were collected by measuring electrolytes before and after applying reduction potential (0 V vs. RHE) flowing $O_2(g)$ or $O_2(g)$+$CH_4(g)$ under the addition of DMPO (Fig. 2b). We confirmed that no radical signal was observed in the electrolyte before the reaction (gray line). Meanwhile, a quartet spectrum with a relative intensity ratio of 1:2:2:1 appeared after applying potential under $O_2$ + $CH_4$ feeding (blue line). It is

typically attributed to the formation of DMPO−OH adducts supporting that radicals were formed during the EMPO reaction[39]. Especially, DMPO−OH adducts were also detected when only $O_2$ was fed without $CH_4$ suggesting that ROSs are generated as the result of electrochemical ORR in our system (red line). Although the signal corresponding to DMPO−OH adducts was the only EPR signal detected, the formation of other ROSs cannot be excluded because other DMPO−ROS adducts have a short lifetime and quickly decompose to the DMPO−OH adduct[40]. Therefore, we further performed trapping experiments applying tert-butyl alcohol (TBA) and 1,4-benzoquinone (BQ) as •OH and •OOH radical scavengers, respectively. To identify the effects of the ROSs, we measured the product amount of $CH_3OOH$ in the presence or absence of the radical scavengers. No $CH_3OOH$ was detected under an excess amount (400 μmol) of scavengers (Fig. 2c). When 40 μmol of TBA and BQ were present in the electrolytes, respectively, the generated amount of $CH_3OOH$ was decreased compared to the case without these scavengers (Supplementary Fig. 12). These series results indicate that ROSs generated from $O_2$ activate $CH_4$ to produce $CH_3OOH$.

Then, the reaction temperature was increased to 70 °C to activate c-$H_2O_2$. At 70 °C, the c-$H_2O_2$ reacted with $CH_4$, but only unstable $CH_3OOH$ was produced at 0.8 μmol h$^{-1}$ production rate (Fig. 2a and Supplementary Fig. 13), which was much lower production than that of the EMPO system at room temperature. Other reaction temperatures (50–90 °C) were also tested using c-$H_2O_2$ and all showed only a small amount of $CH_3OOH$ production (Supplementary Figs. 13, 14). The production rate of the unstable $CH_3OOH$ rather decreased as the reaction temperature increased from 70 °C to 90 °C. This supports that the electro-assisted system (e-$H_2O_2$) contributes to the increase in liquid products. Based on these results, we speculated that the EMPO system not only guides the reaction between $CH_4$ and

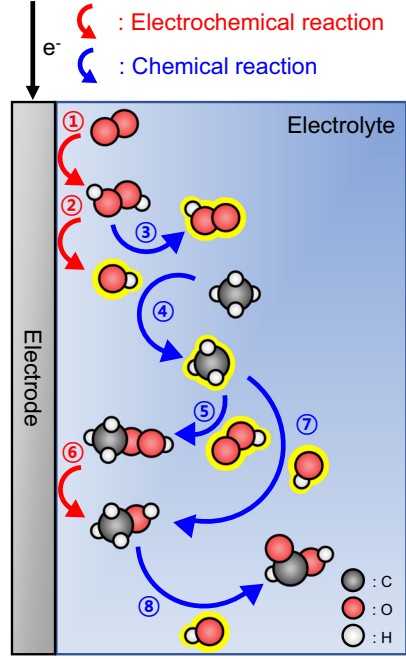

ROSs formation:
$$O_2 + 2H^+ + 2e^- \rightarrow H_2O_2 \qquad ①$$
$$H_2O_2 + e^- \rightarrow \cdot OH + OH^- \qquad ②$$
$$\cdot OH + H_2O_2 \rightarrow \cdot OOH + H_2O \qquad ③$$

$CH_4$ activation:
$$CH_4 + \cdot OH \rightarrow CH_3\cdot + H_2O \qquad ④$$

$CH_3OOH$ formation:
$$CH_3\cdot + \cdot OOH \rightarrow CH_3OOH \qquad ⑤$$

$CH_3OH$ formation:
$$CH_3OOH + 2H^+ + 2e^- \rightarrow CH_3OH \qquad ⑥$$
$$CH_3\cdot + \cdot OH \rightarrow CH_3OH \qquad ⑦$$

HCOOH formation:
$$CH_3OH + 4\cdot OH \rightarrow HCOOH + 3H_2O \qquad ⑧$$

**Fig. 3 | Proposed reaction mechanism of EMPO.** Schematic illustration of the reaction mechanism for selective HCOOH production by EMPO. The red (①,②,⑥) and blue (③,④,⑤,⑦,⑧) arrows and numbers represent electrochemical and chemical reactions, respectively. The black, red, and white balls represent C, O, and H atoms, respectively.

electrochemically generated $H_2O_2$ but also alters the selectivity of the reaction and enhances the production rate.

$$CH_3OOH + 2H^+ + 2e^- \rightarrow CH_3OH + H_2O \quad E^o = 1.70\,V \text{ vs. RHE} \qquad (3)$$

Considering the standard reduction potential of $CH_3OOH$ (Eq. (3)), it can be easily reduced to $CH_3OH$ at the working potential of the EMPO system. We investigated the additional reduction of $CH_3OOH$ on the electrode through control experiments. We conducted chronoamperometry at 0 V (vs. RHE) with alternating supply of $O_2 + Ar$ and $O_2 + CH_4$ gas mixtures for 10 min each (Fig. 2d) to understand whether the supply of $CH_4$ affects the current density of the reaction. While the ORR can occur under $O_2 + Ar$ conditions, under $O_2 + CH_4$ conditions, the generation of oxygenates and their sequential reduction can occur together with the ORR. The current during the $O_2 + CH_4$ supply was always higher than that during the $O_2 + Ar$ supply, and the cathodic current density increased slowly, implying that $CH_4$ induced additional reduction reactions. To determine whether oxygenated products were electrochemically reduced further on the cathode, we performed linear sweep voltammetry to study the reduction capabilities of $CH_3OOH$, $CH_3OH$, and HCOOH at the EMPO working potential using various concentrations (0–40 mM) of these compounds[41,42]. As shown in Fig. 2e, the reduction current increased with increasing $CH_3OOH$ concentration, indicating its easy reducibility. In contrast, the polarization curves of the systems containing $CH_3OH$ and HCOOH exhibited no significant differences with the increase in their concentrations (Supplementary Fig. 15). These results suggest that $CH_3OH$ and HCOOH cannot be reduced on the a-KB cathode under these conditions, and the current increase observed in Fig. 2d is due to the reduction of $CH_3OOH$.

In addition, to confirm whether $CH_3OH$ can be oxidized to HCOOH in the presence of ROSs, we conducted $CH_3OH$ oxidation experiments using c-$H_2O_2$ and e-$H_2O_2$ at room temperature. Similar to $CH_4$, $CH_3OH$ was oxidized to HCOOH only by in-situ generated $H_2O_2$ (e-$H_2O_2$) and no $CH_3OOH$ was produced (Supplementary Figs. 16, 17 and Fig. 2f)[18,35,43,44]. We also investigated the EMPO reaction under the alkaline electrolyte (0.1 M KOH). Although an a-KB exhibited high

selectivity to $H_2O_2$ from ORR in the alkaline condition, no products from $CH_4$ oxidation were observed. This implies that the coupling between ORR and $CH_4$ oxidation reaction is influenced by the pH of the electrolyte, favoring the acidic condition for the EMPO reaction (Supplementary Figs. 18, 19).

### Proposed mechanism
Based on the above electrochemical analyses, we deduced the reaction mechanism of EMPO. As illustrated in Fig. 3, first, $O_2$ is reduced to $H_2O_2$ on the electrode through the electrochemical two-electron pathway ORR, and ROSs are formed (ROSs formation). Second, •OH radicals activate $CH_4$ to produce $CH_3$• radicals in the electrolyte ($CH_4$ activation). Then, $CH_3OOH$ is generated by the reaction between $CH_3$• and •OOH radicals ($CH_3OOH$ formation). Subsequently, $CH_3OH$ was generated by electrochemical reduction of $CH_3OOH$ and radical reaction between $CH_3$• and •OH. Finally, HCOOH was formed in the presence of •OH radicals (HCOOH formation). Thus, HCOOH can be generated as a selective reaction product with the aid of electrochemical reduction potential at room temperature and atmospheric pressure in the EMPO system.

### Electro-assisted $C_2H_6$ partial oxidation
To validate the electro-assisted C−H activation of alkane for partial oxidation products generation, the $C_2H_6$ partial oxidation was examined under the same cathodic $H_2O_2$ generation conditions. All the reaction conditions were maintained the same, except that $C_2H_6$ was used instead of $CH_4$. In this case, $C_2$ oxygenates ($C_2H_5OH$, $C_2H_5OOH$, and $CH_3COOH$) and $C_1$ oxygenates ($CH_3OH$, $CH_3OOH$, and HCOOH) were obtained as the products (Supplementary Fig. 20). The exclusive observation of the $C_2$ partial oxidation products are supporting direct C−H activation from $C_2H_6$. The reaction products were changed from $C_1$ ($CH_3OH$, $CH_3OOH$, and HCOOH) to $C_2$ ($C_2H_5OH$, $C_2H_5OOH$, and $CH_3COOH$) with similar oxidation states by simply replacing $CH_4(g)$ with $C_2H_6(g)$ under the same reaction conditions. The reaction pathway is similar to that of EMPO. As $C_2H_6$ is more reactive ($\Delta H_{C\text{-}H} = 421.7$ kJ mol$^{-1}$)[45,46] than methane, the activation of $C_2H_6$ via C−H bond breakage can occur more easily, resulting in

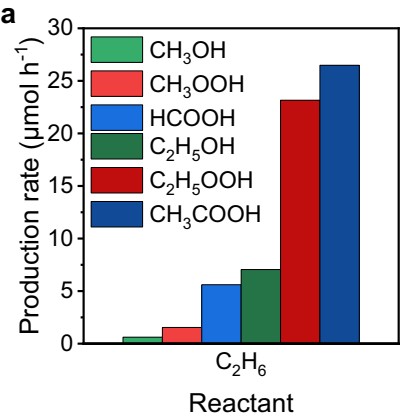

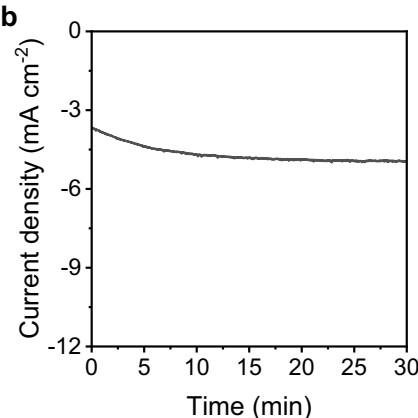

**Fig. 4 | Electro-assisted C$_2$H$_6$ partial oxidation. a** Production rate of electro-assisted ethane partial oxidation. **b** Current density during the reaction time. Reaction conditions: 25 °C, 1 bar, 30 min, O$_2$: 100 sccm, C$_2$H$_6$: 100 sccm, 55 mL of 0.05 M H$_2$SO$_4$, stirring at 700 rpm, and 0 V (vs. RHE). Source data are provided as a Source Data file.

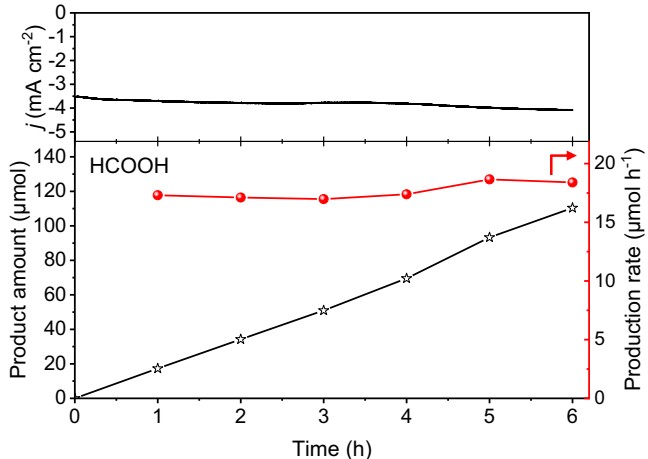

**Fig. 5 | Stability test of the EMPO system.** EMPO reaction during 6 h. Reaction conditions: 25 °C, 1 bar, O$_2$: 100 sccm, CH$_4$: 100 sccm, 550 mL of 0.05 M H$_2$SO$_4$ circulated by 10 mL min$^{-1}$, stirred at 700 rpm, and 0 V (vs. RHE). Source data are provided as a Source Data file.

higher total product yield than that of the EMPO system (Figs. 4a, 2a). Further, C − C bond breakage can lead to C$_1$ oxygenates products. The increased current compared with that of EMPO (Fig. 2a) is attributed to the relatively high concentration of reaction intermediates that can be reduced on the electrode. Through the electro-assisted C$_2$H$_6$ partial oxidation, we confirmed that electrochemically generated H$_2$O$_2$ can also activate C$_2$H$_6$, leading to the production of partially oxidized products. This reaction also demonstrates the general applicability of the electro-assisted system in the partial oxidation of saturated hydrocarbons. Thus, the EMPO system can be adopted as a general partial oxidation system for small hydrocarbons.

### Stability test
A stability test of the EMPO system was performed (Fig. 5). The stability test was conducted under the electrolyte-flowing H-cell setup to keep HCOOH concentration low because the high concentration of HCOOH induces over-oxidation to CO$_2$ (Supplementary Figs. 21, 22). During six hours of the EMPO reaction, no decreases in the current density or production rate of HCOOH were observed demonstrating its stable operation. It is attributed to the stability of metal-free carbon catalyst which operates stably under acidic condition. Additionally, it was

confirmed that no Pt metal impurities were detected in either the catalyst or solution after the reaction by inductively coupled plasma optical emission spectroscopy (ICP-OES) (Supplementary Table 1). The HCOOH production rate in our EMPO process was kept high, and the reaction proceeded under ambient pressure and temperature (Supplementary Fig. 23 and Supplementary Table 2)[7,22,47–54].

## Discussion
In this study, we investigated an EMPO system for the partial oxidation of methane at room temperature and ambient pressure with the assistance of electrochemically in-situ generated H$_2$O$_2$ on the cathode. The EMPO process involves cathodic H$_2$O$_2$ production from the ORR and subsequent partial oxidation of methane, resulting in selective HCOOH production in the acidic condition. Metal-free a-KB was utilized as the catalyst to generate H$_2$O$_2$ via the two-electron pathway ORR. CH$_3$OH, CH$_3$OOH, and HCOOH were obtained as liquid products of EMPO reaction with a total oxygenate production rate of 18.9 μmol h$^{-1}$ under ambient pressure and temperature. Through a mechanistic study of the EMPO process, we propose that ROSs are the active species of the partial oxidation of CH$_4$. CH$_4$ can be activated by the in-situ cathodically generated ROSs (i.e., •OH and •OOH radicals) from the O$_2$ reduction. Additionally, the unstable CH$_3$OOH product can be converted to CH$_3$OH on the cathode at the EMPO working potential; this improved product selectivity (80.7%) toward the stable liquid fuel, HCOOH. This EMPO system applying metal-free carbon catalyst showed significant stable performance for 6 h. We also demonstrated the general applicability of the electro-assisted oxidation process by activating the C−H bond of C$_2$H$_6$. In this case, various C$_2$ and C$_1$ oxygenates were produced, with C$_2$H$_5$OOH and CH$_3$COOH as the major products. We believe that the EMPO system is a promising route for developing a sustainable and miniaturized strategy for on-site natural gas conversion.

## Methods
### Acid treated ketjen black (a-KB) preparation
Ketjen black (EC-600JD) (KB) powder (1 g) was added to 250 mL of 60% HNO$_3$ (Samchun Chemicals) then the mixture was stirred at 80 °C in an oil bath for 12 h. After acid-treatment, acid treated ketjen black (a-KB) was vacuum-filtered and washed with copious amount of deionized (DI) water and dried in vacuum oven overnight. Finally, 800 mg of a-KB powder was obtained.

### Characterization
The XPS spectra were acquired using a Nexsa (ThermoFisher Scientific) instrument with a Microfocus monochromatic Al-Kα X-ray source.

The HR-TEM was performed on a JEM-F200 (JEOL) electron microscope with 200 kV acceleration voltage. EDS mapping were analyzed by JEOL Dual SDD system with 200 kV. The ICP-OES was measured by iCAP 7000 (Thermo Scientific) to analyze the presence of Pt metal impurities in the catalyst and electrolytes.

## Electrochemical ORR test

The electrochemical ORR test was performed using an electrochemical workstation (CHI760E, CH Instruments) at room temperature (25 °C) under atmospheric pressure. For the rotating ring disk electrode (RRDE) measurements, a three-electrode system was constructed with an RRDE (Pine research, E7R9 RRDE) (glassy carbon (GC) disk (0.2475 cm$^2$) + Pt ring (0.1866 cm$^2$)), a Ag/AgCl (stored in 3 M KCl, BASi) reference electrode, and a Pt foil counter electrode. The catalyst ink were prepared by dispersing the a-KB powder in solution (Ethanol ($C_2H_5OH$, Sigma-Aldrich, 99.5%):DI = 3.5:1) with 5 wt% Nafion (Sigma-Aldrich) to achieve concentration of ~0.01 mg µl$^{-1}$. After sonication for 30 min, 8 µl of the catalyst ink was drop casted onto a disk electrode and dried at room temperature. Cyclic voltammetry (CV) was performed between 0.05 and 1.20 V (vs. RHE) in $N_2$ (99.999%)-saturated 0.05 M $H_2SO_4$ (Sigma-Aldrich, 99.999%) and 0.1 M KOH (Sigma-Aldrich, 90%) at a scan rate of 100 mV s$^{-1}$ for 10 cycles, in which steady CV response was obtained. $O_2$ (99.999%) gas was supplied into the electrolyte for 5 min. The impedance spectroscopy was conducted at 0.68 V (vs. RHE) from 100,000 to 1 Hz to determine the uncompensated resistance ($R_u$) in a high-frequency range for $iR$-correction. The $H_2O_2$ production activity was assessed by linear sweep voltammetry (LSV) from 1.1 to 0.2 V (vs. RHE) in $O_2$-saturated 0.05 M $H_2SO_4$ and 0.1 M KOH at a scan rate of 5 mV s$^{-1}$ and rotating speed of 1600 rpm. During the LSV, the Pt ring potential was held at 1.23 V (vs. RHE). The $H_2O_2$ selectivity was calculated using the following relation:

$$H_2O_2 \ Selectivity \ (\%) = 200 \times \frac{i_r/N}{i_d + i_r/N} \quad (4)$$

where $i_r$, $N$, and $i_d$ denote the ring current, collection efficiency (37%), and disk current, respectively.

The Faradaic efficiency of $H_2O_2$ was calculated by equation below:

$$F.E._{H_2O_2}(\%) = 100 \times \frac{i_r/N}{i_d} \quad (5)$$

The collection efficiency ($N$) was determined using the $[Fe(CN)_6]^{3-/4-}$ redox system. Chronoamperometry was carried out at −0.3 V (vs. Ag/AgCl) while the ring potential was fixed at 0.5 V (vs. Ag/AgCl) for 60 s on the catalyst-deposited RRDE in $N_2$-saturated 0.1 M KOH + 2 mM $K_3[Fe(CN)_6]$ (Sigma-Aldrich, ≥99.0%). The background current was obtained similarly, but the disk potential was 0.5 V (vs. Ag/AgCl). The collection efficiency was calculated as follows:

$$N = \frac{|i_r - i_{r,bg}|}{i_d} \quad (6)$$

Where $i_r$, $i_{r,bg}$, and $i_d$ denote the ring current, background ring current, and disk current, respectively. The collection efficiency was 37% (Supplementary Fig. 24).

## Working electrode preparation

5 mg of a-KB catalyst was mixed with 1 mL of 2-propanol (IPA, Sigma Aldrich, 99.5%) and 50 µL of 5 wt% Nafion. Then the mixture was sonicated for 10 min. The catalyst ink was sprayed on the carbon paper (TGP-H-120 20%WP, Toray) (11.25 cm$^2$). The mass loading of a-KB on the carbon paper was 1 mg cm$^{-2}$.

## EMPO experiment

The EMPO experiment was conducted in an H-type cell separated by Nafion 117 membrane using a potentiostat (Ivium Vertex, Ivium Technologies). A three-electrode system consisting of an a-KB sprayed carbon paper as the working electrode, Ag/AgCl (stored in 3 M KCl) reference electrode, and a Pt plate as a counter electrode was constructed. The working electrode was used without pre-activation process. The 0.05 M $H_2SO_4$ was used as an electrolyte (55 mL for both catholyte and anolyte). Catholyte was purged with $O_2$ and $CH_4$ (99.999%) gases for at least 30 min prior to reaction and constantly purged at a flow rate of 100 sccm during the reaction (0 V vs. RHE). The catholyte was continuously stirred at 700 rpm. Reaction temperature was maintained at 25 °C by water bath.

For the control experiment, all reaction conditions were the same except for feeding Ar gas (99.999%) instead of $CH_4$.

For the $^{13}CH_4$ (Icon Isotopes, 99.8%) isotope experiment, all reaction conditions were the same except for feeding $^{13}CH_4$ instead of $^{12}CH_4$.

For the carbon-free ORR catalyst experiment, Au foil (Dasom RMS, 99.99%, 11.25 cm$^2$) was applied for working electrode while all other reaction conditions were the same. The Au foil was used without pre-activation process.

For the time dependent production rate analysis, potential (0 V vs. RHE) was applied for 1 h and 500 µL of electrolyte was extracted for each 5, 10, 15, 20, 25, 30, and 60 min. Other reaction conditions were the same.

For the flow rate experiments, the flow rates of $CH_4$ and $O_2$ were varied from 25 sccm to 200 sccm while all other reaction conditions were maintained.

For the potential dependent production rate analysis, applied potential was varied from −0.2 to +0.2 V (vs. RHE) for 30 min. Other reaction conditions were the same.

For the identification of O source experiment, Ar was purged instead of $O_2$. Other reaction conditions were the same.

For the EPR experiment, 1 mmol of DMPO (Sigma-Aldrich, ≥98%) was applied as a spin trap before the reaction. Other reaction conditions were the same except for feeding $O_2$ or $O_2 + CH_4$ gases.

For the trapping experiments, 400 and 40 µmol of TBA (Sigma-Aldrich, ≥99.5%) and BQ (Sigma-Aldrich, ≥98%) were added before the reaction as •OH and •OOH radical scavengers, respectively. Other reaction conditions were the same.

For the chronoamperometry experiment, $O_2 + Ar$ and $O_2 + CH_4$ (100 sccm each) were purged alternatively every 10 min for 1 h. Other reaction conditions were the same.

For the $CH_3OH$ (Sigma-Aldrich, ≥99.9%) oxidation experiment, all reaction conditions were the same except for injecting 100 µmol of $CH_3OH$ before the reaction instead of $CH_4$.

For the alkaline EMPO experiment, all reaction conditions were the same except for applying 0.1 M KOH as an electrolyte instead of 0.05 M $H_2SO_4$.

For the electro-assisted $C_2H_6$ partial oxidation experiment, all reaction conditions were the same except feeding $C_2H_6$ instead of $CH_4$.

For the stability test, the electrolyte-flowing H-cell setup was established. 550 mL of 0.05 M $H_2SO_4$ was circulated by 10 mL min$^{-1}$. The EMPO reaction was performed for 6 h. Other reaction conditions were the same.

## Methane partial oxidation using commercial $H_2O_2$

$CH_4$ partial oxidation using commercial $H_2O_2$ (30 wt% in $H_2O$, Sigma-Aldrich) was conducted using cathode part of H-type cell. 55 mL of 0.05 M $H_2SO_4$ was filled in cathode part and the solution was purged with Ar and $CH_4$ at 100 sccm simultaneously. Ar was purged instead of $O_2$ to keep balance of partial flow of $CH_4$. Commercial $H_2O_2$ was injected during reaction time (30 min) via syringe pump. The amount

of $H_2O_2$ supplied was determined based on RRDE analysis.

$$The\ amount\ of\ H_2O_2 = \frac{F.E._{\cdot H_2O_2} \times j \times A \times t}{n \times F} \qquad (7)$$

where, $F.E._{\cdot H2O2} = 81.8\%$, $j$ is current density, $A$ is area of electrode, $t$ is reaction time, $n$ is moles of electron to produce 1 mole of $H_2O_2$, and $F$ is faraday constant. The solution was stirred at 700 rpm and kept at 25, 50, 70, or 90 °C by water bath.

## Methanol oxidation using commercial $H_2O_2$

$CH_3OH$ oxidation using commercial $H_2O_2$ was conducted using cathode part of H-type cell. 55 mL of 0.05 M $H_2SO_4$ was filled in cathode part and 100 μmol of $CH_3OH$ was injected before the reaction. Commercial $H_2O_2$ was fed during reaction time (30 min) via syringe pump. The amount of supplied $H_2O_2$ was determined based on RRDE analysis.

## ¹H-NMR analysis of liquid products

The concentration of the liquid products was quantified by ¹H-NMR (Agilent 600 MHz). Typically, 430 μL sample was mixed with 50 μL of $D_2O$, and 20 μL of 6 mM dimethyl sulfoxide (DMSO, Sigma-Aldrich, 99.9%) was added as an internal standard.

## ¹³C-NMR analysis of liquid products

Due to the lower detection limit of ¹³C-NMR than that of ¹H-NMR, freeze-drying prior to analysis to combat low concentrations, frequently <100 μM C, was conducted[55]. 5 mL of 1 M KOH was mixed with 50 mL of sample to alkalify formic acid to formate anion to prevent sublimation during freeze-drying. Then sample was frozen by liquid nitrogen. Frozen sample was lyophilized at −80 °C at 200 mtorr until dry (48 h) ($CH_3OH$ and $CH_3OOH$ was sublimated in this step). Freeze-dried sample was re-dissolved in 1 mL DI water. 500 μL of re-dissolved sample was qualified by ¹³C-NMR (Agilent 600 MHz).

## GC analysis of gas products

The gas products from the EMPO reaction were quantified by on-line gas chromatography (GC, Agilent 6890). A flame ionization detector was used to detect CO and $CO_2$. A methanizer was utilized to increase the detection sensitivity of CO and $CO_2$. The GC system was equipped with a ShinCarbon ST column (Restek) to separate gas products. The calibration of the GC was carried out by flowing three calibration gas mixtures with CO- and $CO_2$-concentrations ranging from 10 to 100 ppm (Supplementary Fig. 25).

## EPR experiment

The EPR spectrum of electrolytes was measured at KBSI Seoul Western Center using CW/Pulse EPR system with the following parameters: frequency 9.852 GHz; power 3 mW; modulation frequency 100 kHz; modulation amplitude 1 G; time constant 20.48 ms; conversion time 20.00 ms; scan 8; room temperature.

## Synthesis of $CH_3OOH$

The $CH_3OOH$ was synthesized by the modified procedures from the previously-reported method[56]. On the three-neck flask, 62.5 g of $H_2O$, 37.5 g of 30% $H_2O_2$, and 25 g of dimethyl sulfate (($CH_3O)_2SO_2$, Sigma-Aldrich, ≥99.5%) were mixed. Subsequently, 52.5 g of 40 wt% KOH aqueous solution was added dropwise very slowly into the solution with stirring to induce a nucleophilic substitution reaction. The overall reaction is exothermic. Then, the gas-phase products were generated by the heat of the reaction. The generated gas-phase products were collected as a liquid phase in the vial by the condensation. Diluting the concentrated $CH_3OOH$ in the DI water, electrochemical analysis was performed. The synthesized $CH_3OOH$ was confirmed and quantified by ¹H-NMR analysis (Supplementary Fig. 26).

## Reduction reaction test of $CH_3OOH$, $CH_3OH$, and HCOOH

The reducibility test of $CH_3OOH$, $CH_3OH$, and HCOOH (Sigma-Aldrich, ≥98%) were also performed in the similar electrochemical configurations as mentioned above. A working electrode prepared by spraying a-KB based catalyst ink on carbon paper (11.25 cm²) was used. The mass loading of a-KB on the carbon paper was 1 mg cm⁻². Before the test, CV was carried out for the electrochemical cleaning of the electrode surface between 0.05 and 1.2 V (vs. RHE) at a scan rate of 100 mV s⁻¹ for 20 cycles in $N_2$-saturated 0.05 M $H_2SO_4$. LSV was then performed from 1.1 to 0.1 V (vs. RHE) at a scan rate of 10 mV s⁻¹ in $N_2$-saturated 0.05 M $H_2SO_4$ containing 0, 10, 20, and 40 mM of $CH_3OOH$, $CH_3OH$, and HCOOH, respectively.

## Calculation of Gibbs free energy and standard potential

Standard Gibbs free energies of selected coupled reactions and their corresponding standard cell potentials have been calculated by Eqs. (8, 9) based on thermodynamic data (Supplementary Table. 3)[57]

$$\triangle G = \triangle H - T \triangle S \qquad (8)$$

$$E = \frac{-\triangle G}{nF} \qquad (9)$$

## Data availability

The authors declare that the data supporting the findings of this study are available within the article and its Supplementary Information files. Source data are provided as a Source Data file. Additional data are available from the corresponding author upon reasonable request. Source data are provided with this paper.

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

## Acknowledgements

This work was supported by Yonsei-KIST Convergence Research Program (J.H.P) and Yonsei Fellow Program, funded by Lee Youn Jae (J. K.). The National Research Foundation of Korea (NRF) grant funded by the Korea government (NRF–2021R1A5A1084921 and 2021M3H4A1A03057403 (Y.J.H.), NRF-2019R1A2C3010479 (J.H.P.)), the Institute for Basic Science (IBS-R006-D1; Y.J.H.), and Creative-Pioneering Researchers Program through Seoul National University also support this work (Y.J.H.).

## Author contributions

Y.J.H. and J.H.P. conceived and supervised the project. J.K., J.H.K., J.H.P., and Y.J.H. prepared the manuscript. J.H.K. and J.K. synthesized the electrocatalyst and performed electrochemical experiments. J.H.K. synthesized $CH_3OOH$ and E.L. assisted synthesis process. J.K. and H.Y. conducted characterization of the electrocatalyst. J.K. and C.O. performed NMR analysis. C.O., H.O., J.H.P., and Y.J.H. provided advice and expertise. All authors discussed the result and commented on the manuscripts.

## Competing interests

The authors declare no competing interests.
