## [Peer Review File · Nature Communications]

Electro-assisted methane oxidation to formic acid via in-situ cathodically generated H₂O₂ under ambient conditionsREVIEWER COMMENTS

Reviewer #1 (Remarks to the Author):

This paper presents a study of electro-assisted CH₄ oxidation to formic acid using in situ generated H₂O₂ oxidant under ambient conditions. Although the idea is not completely novel, the results are interesting yet presented in a brief narration and experimental findings. The mechanistic study lacks strong experimental proofs. Also, including C₂H₆ oxidation does not sound a convincing approach for understanding CH₄ oxidation mechanism. Some specific comments are provided below.

1. Why are the authors sure that H₂O₂ (not other intermediate active oxygen species) is oxidizing CH₄ at room temperature? If that is the case, then CH₄ can simply be oxidized by bubbling it in an electrolyte with H₂O₂. However, the results shown in Fig. 2a suggest a different conclusion where no liquid products were generated when commercial H₂O₂ has been used at room temperature. The authors attributed this to the presence of stabilizers that inhibit the decomposition of H₂O₂. But, if the reaction takes place chemically between CH₄ and H₂O₂, then why do we need H₂O₂ to decompose? Heating the system to 70 degrees forces H₂O₂ to decompose and form ROSs which will attack CH₄ to form CH₃OOH. A report by Jin et al. (Science 367, 193–197 (2020)) showed much better performance at reaction temperatures as low as 70
2. Similarly, there is not enough evidence that the generated CH₃OH from electrochemical reduction of CH₃OOH is oxidized by H₂O₂ at room temperature. The CH₃OH oxidation experiment presented in Fig. 2c has been performed in the same condition of the ORR experiment.
3. It is hard to digest that the EMPO reaction of CH₄ to formic acid is smoothly switching back and forth between surface catalysis and bulk chemical reaction as summarized in Fig. 3. A control experiment or literature-backed discussion is needed to be provided to justify this mechanism.
4. What is the reason behind using an acidic electrolyte (0.05 M H₂SO₄)?
5. (minor) Fig. 2c discussed after Figs 2d-f. Please rearrange figures order.
6. Reaction mechanism discussed/elaborated in lines 293-243 needs to be revised according to the comments mentioned above.
7. The results shown under the "Electro-assisted C₂H₆ partial oxidation" section are difficult to be related to CH₄. C-H activation of CH₄ is significantly more challenging than that of C₂H₆. The authors need to provide more convincing reasoning for including C₂H₆ oxidation in this study.
8. Cycle stability experiment is understandable for photo(electro)-catalysis. However, ORR experiments (and electrocatalysis studies in general) are usually presented in one run over long operation time to comment on the catalyst as well as overall system stability.
9. Were there any gaseous products from the EMPO? CO₂ is expected to be produced from inevitable over-oxidation of CH₄.
10. How was the 83.6% selectivity of HCOOH calculated? And does selectivity here includes all products or limited to liquid products (liquid product selectivity).
11. (minor) in line 27 in the abstract "Here, we present a novel approach...". The approach is not novel as mentioned in lines 90-94. So, please rewrite this sentence.

Reviewer #2 (Remarks to the Author):

In the manuscript, Jong Hyeok Park et al reported an approach for the partial oxidation of methane, viz., electro-assisted methane partial oxidation (EMPO) using in-situ cathodically generated H₂O₂, which can be accomplished at ambient pressure and temperature. Upon using activated carbon as the electrocatalyst, the EMPO process enables the partial oxidation of methane in an acidic electrolyte to produce oxygenated liquid products. However, the mechanism and some characterizations of this paper are unclear enough. The authors should consider the following specific comments in a possible revision and submit the revised version to more specialized journals.

Specific comments:

1. The H₂O₂ produced in situ does not have the ability to activate the methane C-H bond, so it needs to be activated to produce active species, such as ·OH. However, this paper did not define the active species, and the explanation of H₂O₂ activation mechanism and methane activation mechanism was insufficient.
2. Supplementary Figure 7 is not described in the text. Please complete it.
3. There is a mixture of upper and lower case in the title of references. Please carefully check the format of the reference document and modify it.
4. Some tenses are inaccurate, should use simple past tense, please carefully check the tenses of the whole text and carefully modify.

Reviewer #3 (Remarks to the Author):

In the submission the authors proposed an electro-assisted methane partial oxidation (EMPO) approach for the partial oxidation at ambient pressure and temperature using in-situ cathodically generated H₂O₂. Using activated carbon as the electrocatalyst, the EMPO process enabled the partial oxidation of methane in an acidic electrolyte to selectively produce formic acid (83.6%). A reaction mechanism with unstable methyl peroxide generated from methane partial oxidation as the intermediate was proposed. Such an electro-assisted oxidation process was also demonstrated effective for ethane partial oxidation.

The reaction data are interesting and the results are presented in a clear way, but the mechanistic studies are not sufficient. The following issues must be clarified before the submission can be accepted:

1. Were CO and CO₂ produced during the catalytic reaction? How about the carbon balance?
2. The authors used activated carbon as the electrocatalyst. Although ¹³C-CH₄ was used to demonstrate that the products came from CH₄, it could not be excluded that the carbon atoms in the catalyst participated into the reaction because the chemical bonds in the activated carbon are weaker than that in CH₄. Is it possible to use non-carbon-containing catalysts to test the idea?
3. The authors proposed in-situ cathodically generated H₂O₂ as the reactive species, however, is it possible that the more reactive OOH species, a likely intermediate during H₂O₂ production, directly acted as the active species?
4. In the controlled c-H₂O₂ system experiments, c-H₂O₂ was supplied with a syringe pump and injected at the same rate as the H₂O₂ production rate in the e-H₂O₂ system, and no liquid products were generated. The authors proposed that the stabilizers contained in c-H₂O₂ (stannate- and phosphorus-containing compounds) inhibited the decomposition of H₂O₂. Where did the proposed stabilizers come from? And what happened if the H₂O₂ supply was increased?
5. Where did the O atoms in the products come from, O₂ or H₂O?
6. What were the advantages of the activated carbon electrocatalyst and how did it work?
7. The stability test seemed for the electrocatalyst stability, out of the focus of the study? The authors need to develop a flow reactor for the EMPO approach?
8. In connection to Comment 7: More parameters, such as CH₄ and O₂ flow rates, and CH₄:O₂ ratios,

need to be examined on the reaction? Do the solubility of CH₄ and O₂ affect the reaction?

Reviewer #4 (Remarks to the Author):

In the submitted manuscript, the authors developed an electrochemical system to valorize methane into formate as a liquid value-added chemical. They carried out methane conversion by first reducing oxygen to peroxide via a carbon catalyst, which then reacted with CH₄ in a chemical step to generate CH₃OOH. This species was then reduced to CH₃OH electrochemically and oxidized chemically again by the peroxide to yield HCOOH as the final product.

The system is rather simple and proposed mechanism quite reasonable and supported with an intuitive set of experiments. This is a significant value-added as electrochemical methane oxidation systems are not well established and the few that do exist often require harsh conditions and feature low yields. Because of this, I would recommend publication in Nature Communications with several suggestions below.

Introduction – would be worth mentioning the carbon footprint of steam reforming vs electrochemical valorization of methane.

Why is such a large overpotential required for ORR? State-of-the-art carbon-based catalysts can catalyze this reaction near the thermodynamic onset potential of approx. 0.7V vs. RHE.

The collection efficiency of 37% (Page 19, line 331) is higher than a typical collection efficiency of 25%, based on the reviewer's experience. Please provide a technical and commercial details of the RRDE electrode. If it is possible, please provide a collection efficiency measurement in the supplementary information.

It seems the authors used molar fraction selectivity and faradaic efficiency interchangeably. The H₂O₂ selectivity formula in page 19 is based on the molar fraction selectivity, which is always higher than faradaic efficiency. (Nature Catalysis, 2020, 3, 605–607). The reported selectivity of 90% in Supplementary Figure 3, is the same with FE mentioned in Page 21 line 367. Please keep consistent.

Page 13 line 214 mentions "These results suggest that CH₃OH and HCOOH cannot be reduced on the a-KB cathode under these conditions, and the current increase observed in Fig. 2b is due to the reduction of CH₃OOH." On the other hand, figure 2f shows that the increasing HCOOH concentration enhances the reduction current, though to a smaller extent. What is occurring here?

It seems the data in Figure 2d, 2e, and 2f were recorded by RRDE. If so, please provide the ring current on the figures in supplementary information.

Please have a look at "ACS Catalysis 2018, 8, 9, 7961–7972", arguing peroxide assisted methane oxidation via a Fenton reaction. In there, CH₃OH formation is considered to happen in parallel to CH₃OOH production, with multiple possible paths. No formic acid was observed. It is better to explain the difference between that article and this study.

Responses to Reviewers' comments

We appreciate the reviewers for the valuable comments which improve our manuscript. We revised the manuscript thoroughly based on the comments, and the changes in the revised manuscript are highlighted in red.

Reviewer(s)' Comments to Author:

Reviewer #1: This paper presents a study of electro-assisted CH₄ oxidation to formic acid using in situ generated H₂O₂ oxidant under ambient conditions. Although the idea is not completely novel, the results are interesting yet presented in a brief narration and experimental findings. The mechanistic study lacks strong experimental proofs. Also, including C₂H₆ oxidation does not sound a convincing approach for understanding CH₄ oxidation mechanism. Some specific comments are provided below.

Response: Thank you for your positive comments on our manuscript, and we understand the reviewers' concern on the mechanistic study. Therefore, in the revised manuscript, we focus on additional experiments that can provide more direct mechanistic information on whether reactive oxygen species are involved during the electrochemical CH₄ oxidation reaction. With the inclusion of these new results and findings, it is hoped that the revised manuscript is a more comprehensive and convincing account of the mechanistic insights. The detailed revisions are listed below. Thank you for your valuable comments again.

1. Why are the authors sure that H₂O₂ (not other intermediate active oxygen species) is oxidizing CH₄ at room temperature? If that is the case, then CH₄ can simply be oxidized by bubbling it in an electrolyte with H₂O₂. However, the results shown in Fig. 2a suggest a different conclusion where no liquid products were generated when commercial H₂O₂ has been used at room temperature. The authors attributed this to the presence of stabilizers that inhibit the decomposition of H₂O₂. But, if the reaction takes place chemically between CH₄ and H₂O₂, then why do we need H₂O₂ to decompose? Heating the system to 70 degrees forces H₂O₂ to decompose and form ROSs which will attack CH₄ to form CH₃OOH. A report by Jin et al. (Science 367, 193-197 (2020)) showed much better performance at reaction temperatures as low as 70.

Response: Thank you for your valuable comments to raise the question whether the active oxygen species are oxidizing CH₄. We also agree with the reviewer's points that CH₄ can be oxidized by the reactive oxygen species (ROSs) rather than direct H₂O₂ molecule itself. It is plausible that heating the system can force it to decompose H₂O₂ and form ROSs. Also, ROSs can be formed during electrochemical oxygen reduction reaction (ORR) (*Angew. Chem. Int. Ed.* **60**, 10375-10383 (2021) / *J. Phys. Chem. Lett.* **12**, 7797-7803 (2021)). Therefore, we performed experiments to figure out whether ROSs are formed and participate in the EMPO reaction.

First, we conducted trapping experiments applying tert-butyl alcohol (TBA) and 1,4-benzoquinone (BQ) as ·OH and ·OOH radical scavengers, respectively. If these ROSs are generated and participate in EMPO reaction, the amount of reaction products should be affected in the presence of radical scavengers because ROSs can be trapped. Because CH₃OOH, which contains a methyl group (CH₃-), can be formed directly through the partial oxidation of CH₄, we measured the product amount of CH₃OOH in the presence or absence of the radical scavengers. As you can see in Figure R1, no CH₃OOH was detected under an excess amount (400 μmol) of scavengers. In addition, when 40 μmol of TBA and BQ were present in the electrolytes, respectively, the generated amount of CH₃OOH was decreased significantly compared to the case without these scavengers. These series results indicate that these radical processes are involved in CH₃OOH formation from CH₄ oxidation.

Figure R1. CH₃OOH production after EMPO reaction in the presence of a) 400 μmol and b) 40 μmol of scavengers. TBA and BQ are radical scavengers that can trap ·OH and ·OOH, respectively.

Next, to verify radical species more directly, we additionally conducted electron paramagnetic resonance (EPR) analysis by adding 5,5-dimethyl-1-pyrroline N-oxide (DMPO) in the electrolytes as a spin trap during EMPO reaction. EPR signals were collected by measuring electrolytes before and after applying reduction potential (0 V vs. RHE) flowing $O_2(g)$ or $O_2(g)+CH_4(g)$ under the addition of DMPO (Figure R2). We confirmed that no radical signal was observed in the electrolyte before the reaction (grey line in Figure R2). Meanwhile, a quartet spectrum with a relative intensity ratio of 1:2:2:1 appeared after applying potential under O_2+CH_4 feeding (blue line). It is typically attributed to the formation of DMPO–OH adducts (*Free Radic. Biol. Med.* **3**, 259-303 (1987)) supporting that radicals were formed during EMPO reaction. Especially, DMPO–OH adducts were also detected when only O_2 was fed without CH_4 suggesting that ROSs are generated as the result of electrochemical ORR in our system. Although the signals corresponding to DMPO–OH adducts was the only EPR signal detected, the formation of other ROSs cannot be excluded because other DMPO–ROS adducts have a short lifetime and quickly decompose to the DMPO–OH adduct (*Free Radic. Res. Commun.* **19**, S79-S87 (1993)).

Figure R2. EPR spectra of the electrolytes before (grey line) and after applying potential (0 V vs. RHE) under $O_2(g)$ (red line) and $O_2(g)+CH_4(g)$ (blue line) flowing with DMPO.

From these above experiments, we propose that ROSs are formed during the EMPO reaction and they activate CH_4 generating CH_3OOH . These results and detailed experimental methods are now included in the revised manuscript (page 3 line 13-15, page 7 line 1-2, page 12 line 19-page 13 line 5, page 13 line 9-page 14 line 5, page 20 line 7-8, page 20 line 10-12, page 24 line 7-13, page 26 line 15-19, Fig. 2b-c, and Supplementary Fig. 12) as below. In addition,

based on these results, the revised reaction mechanism is discussed in the response to comment #3. Thank you for your comment.

[page 3 line 13-15] The mechanistic study revealed that reactive oxygen species (ROSs) such as $\cdot\text{OH}$ and $\cdot\text{OOH}$ radicals are produced during the reaction and activated methane and methanol.

[page 7 line 1-2] Reactive oxygen species (ROSs) (i.e., $\cdot\text{OH}$ and $\cdot\text{OOH}$ radicals) were generated during ORR and activated CH_4 and CH_3OH .

[page 12 line 19-page 13 line 5] In contrast with EMPO, no liquid products were formed when $c\text{-H}_2\text{O}_2$ was added in the reaction cell at $25\text{ }^\circ\text{C}$ (Fig. 2a and Supplementary Fig. 9-10) implying that H_2O_2 molecule itself cannot react with CH_4 and ROSs formed during electrochemical ORR are crucial for the EMPO.^{37, 38}

We performed control experiments to figure out whether ROSs are generated from O_2 and participate in the EMPO reaction.

[page 13 line 9-page 14 line 5] Then, to verify radical species more directly, we additionally conducted electron paramagnetic resonance (EPR) analysis by adding 5,5-dimethyl-1-pyrroline N-oxide (DMPO) in the electrolytes as a spin trap. EPR signals were collected by measuring electrolytes before and after applying reduction potential (0 V vs. RHE) flowing $\text{O}_2(\text{g})$ or $\text{O}_2(\text{g})+\text{CH}_4(\text{g})$ under the addition of DMPO (Fig. 2b). We confirmed that no radical signal was observed in the electrolyte before the reaction (grey line). Meanwhile, a quartet spectrum with a relative intensity ratio of 1:2:2:1 appeared after applying potential under O_2+CH_4 feeding (blue line). It is typically attributed to the formation of DMPO–OH adducts supporting that radicals were formed during EMPO reaction.³⁹ Especially, DMPO–OH adducts were also detected when only O_2 was fed without CH_4 suggesting that ROSs are generated as the result of electrochemical ORR in our system (red line). Although the signal corresponding to DMPO–OH adducts was the only EPR signal detected, the formation of other ROSs cannot be excluded because other DMPO–ROS adducts have a short lifetime and quickly decompose to the DMPO–OH adduct.⁴⁰ In addition, we further performed trapping experiments applying tert-butyl alcohol (TBA) and 1,4-benzoquinone (BQ) as $\cdot\text{OH}$ and $\cdot\text{OOH}$ radical scavengers, respectively. To identify the effects of the ROSs, we measured the product amount of CH_3OOH in the presence or absence of the radical scavengers. No CH_3OOH was detected under an excess amount (400 μmol) of scavengers (Fig. 2c). When 40 μmol of TBA and BQ were present in the electrolytes, respectively, the generated amount of CH_3OOH was

decreased significantly compared to the case without these scavengers (Supplementary Fig. 12). These series results indicate that ROSs generated from O_2 activate CH_4 to produce CH_3OOH .

[page 20 line 7-8] CH_4 can be activated by the in-situ cathodically generated ROSs (i.e., $\cdot OH$ and $\cdot OOH$ radicals) from the O_2 reduction.

[page 20 line 10-12] Through a mechanistic study of the EMPO process, we propose that ROSs are the active species of the partial oxidation of CH_4 .

[page 24 line 7-13] For the EPR experiment, 1mmol of DMPO (Sigma-Aldrich) was applied as a spin trap before the reaction. Other reaction conditions were the same except for feeding O_2 or O_2+CH_4 gases.

For the trapping experiments, 400 and 40 μmol of TBA (Sigma-Aldrich, $\geq 99.5\%$) and BQ (Sigma-Aldrich, $\geq 98\%$) were added before the reaction as $\cdot OH$ and $\cdot OOH$ radical scavengers, respectively. Other reaction conditions were the same.

[page 26, line 15-19] **EPR experiment**

The EPR spectrum of electrolytes was measured at KBSI Seoul Western Center using CW/Pulse EPR system with the following parameters: frequency 9.852 GHz; power 3 mW; modulation amplitude 1 G; time constant 20.48 ms; conversion time 20.00 ms; scan 8; temperature RT.

2. Similarly, there is not enough evidence that the generated CH_3OH from electrochemical reduction of CH_3OOH is oxidized by H_2O_2 at room temperature. The CH_3OH oxidation experiment presented in Fig. 2c has been performed in the same condition of the ORR experiment.

Response: Thank you for your comment. We also agree with your opinion that there is not enough evidence to show how CH_3OH is oxidized by H_2O_2 at room temperature. Therefore, we conducted the CH_3OH oxidation experiment applying commercial H_2O_2 to confirm whether it can oxidize CH_3OH at room temperature. As shown in Figure R3, no liquid products were detected in 1H -NMR analysis after reaction between CH_3OH and H_2O_2 molecule. Meanwhile, $HCOOH$ was generated by CH_3OH oxidation in the same condition of the ORR experiment (Fig. 2c). These results also support that *in-situ* electrochemical ORR to H_2O_2 production (i.e., ROSs formed during ORR reaction) is crucial to oxidize CH_3OH to $HCOOH$.

Figure R3. ^1H -NMR analysis after reaction between commercial H_2O_2 and CH_3OH at room temperature. The peaks at 2.71, and 3.35 ppm are attributed to $(\text{CH}_3)_2\text{SO}$ (DMSO), and CH_3OH , respectively.

In the presence of $\cdot\text{OH}$ radicals, CH_3OH can be oxidized to HCOOH (*Angew. Chem. Int. Ed.* **51**, 5129-5133 (2012) / *ACS Catal.* **11**, 6684-6691 (2021)).

Above findings are added to the manuscript (page 3 line 13-15, page 7 line 1-2, page 15 line 7-10, page 16 line 10-11, and Supplementary Fig. 15). Thank you for your comment.

[page 3 line 13-15] *The mechanistic study revealed that reactive oxygen species (ROSs) such as $\cdot\text{OH}$ and $\cdot\text{OOH}$ radicals are produced during the reaction and activated methane and methanol.*

[page 7 line 1-2] *Reactive oxygen species (ROSs) (i.e., $\cdot\text{OH}$ and $\cdot\text{OOH}$ radicals) were generated during ORR and activated CH_4 and CH_3OH .*

[page 15 line 7-10] *In addition, to confirm whether CH_3OH can be oxidized to HCOOH in the presence of ROSs, we conducted CH_3OH oxidation experiments using $c\text{-H}_2\text{O}_2$ and $e\text{-H}_2\text{O}_2$ at room temperature. Similar to CH_4 , CH_3OH was oxidized to HCOOH only by in-situ generated H_2O_2 ($e\text{-H}_2\text{O}_2$) and no CH_3OOH was produced (Supplementary Fig. 15-16 and Fig. 2f).^{18, 35, 43}*

44

[page 16 line 10-11] *Finally, HCOOH was formed in the presence of $\cdot\text{OH}$ radicals (HCOOH formation).*

3. It is hard to digest that the EMPO reaction of CH₄ to formic acid is smoothly switching back and forth between surface catalysis and bulk chemical reaction as summarized in Fig. 3. A control experiment or literature-based discussion is needed to be provided to justify this mechanism.

Response: Thank you for the important comments. According to your comments, during the revision process, we systematically conducted several control experiments to identify the reaction mechanism better and were able to confirm the electrochemical generation of ROSs during the EMPO reaction. Based on the above-mentioned experimental results (Figure R1~3), we could verify two facts; i) ROSs are generated during the EMPO reaction, ii) ROSs activate CH₄ and CH₃OH. The previous photocatalytic, catalytic reaction, and density functional theory (DFT) simulation studies proposed that radical species lead to activation of methane during the partial oxidation reaction to produce CH₃OH, CH₃OOH, or HCOOH (*Nat. Catal.* **1**, 889-896 (2018) / *Nat. Commun.* **13**, 2930 (2022) / *Chem. Eur. J.* **18**, 15735-15745 (2012) / *ACS Catal.* **11**, 6684-6691 (2021) / *ACS Catal.* **8**, 7961-7972 (2018)). The proposed mechanism of EMPO reaction is now modified to include radical processes in the activation steps of CH₄ or intermediates as follows (Figure R4):

ROSs formation:

CH₄ activation:

CH₃OOH formation:

CH₃OH formation:

HCOOH formation:

Figure R4. Proposed mechanism. Schematic illustration of the reaction mechanism for selective HCOOH production by EMPO.

Above findings are added to the manuscript (page 16 line 5-13 and Fig.3). Thank you for your comment.

[page 16 line 5-13] As illustrated in Fig. 3, first, O_2 is reduced to H_2O_2 on the electrode through the electrochemical two-electron pathway ORR, and ROSs are formed (ROSs formation). Second, $\cdot\text{OH}$ radicals activate CH_4 to produce $\text{CH}_3\cdot$ radicals in the electrolyte (CH_4 activation). Then, CH_3OOH is generated by the reaction between $\text{CH}_3\cdot$ and $\cdot\text{OOH}$ radicals (CH_3OOH formation). Subsequently, CH_3OH was generated by electrochemical reduction of CH_3OOH and radical reaction between $\text{CH}_3\cdot$ and $\cdot\text{OH}$. Finally, HCOOH was formed in the presence of $\cdot\text{OH}$ radicals (HCOOH formation). Thus, HCOOH can be generated as a selective reaction product with the aid of electrochemical reduction potential at room temperature and atmospheric pressure in the EMPO system.

4. What is the reason behind using an acidic electrolyte (0.05 M H_2SO_4)?

Response: Thank you for your comments on the electrolyte condition. We have tested EMPO reaction in both acidic (0.05 M H₂SO₄) and alkaline (0.1 M KOH) electrolytes. First, we screened ORR performance of an a-KB using a rotating ring disk electrode (RRDE). The a-KB catalyst also exhibited high H₂O₂ selectivity in the 0.1 M KOH electrolyte (Figure R5).

Figure R5. Electrochemical ORR performance for H₂O₂ production of a-KB in an alkaline electrolyte (0.1 M KOH).

Then, we carried out the EMPO reaction in 0.1 M KOH with continuous feeding of O₂ and CH₄ gases at 0 V (vs. RHE). While methane partial oxidation products such as CH₃OH, CH₃OOH, and HCOOH were formed in the acidic electrolyte (0.05 M H₂SO₄), no liquid products were generated in 0.1 M KOH electrolyte (Figure R6). These results were consistent with the previous studies that production of ·OH from the electrochemical reduction of H₂O₂ process mainly reported under low pH electrolyte condition. (*Angew. Chem. Int. Ed.* **60**, 10375-10383 (2021), *J. Phys. Chem. Lett.* **12**, 7797-7803 (2021)). Similarly, formation of ROS from H₂O₂ (*i.e.* Fenton reaction) has been mostly demonstrated in the low pH condition. These imply that the coupling between ORR and CH₄ oxidation reaction is influenced by the pH of the electrolyte, favoring the acidic condition for the EMPO reaction.

Figure R6. ^1H -NMR spectrum of the products after EMPO reaction in the alkaline electrolyte (0.1 M KOH). The peaks at 0.17 and 2.71 ppm are attributed to CH_4 and $(\text{CH}_3)_2\text{SO}$ (DMSO), respectively.

Above findings are added to the manuscript (page 15 line 10-15, page 22 line 3-5, page 22 line 7-10, page 24 line 18-19, and Supplementary Fig.17-18). Thank you for your comment.

[page 15 line 10-15] We also investigated the EMPO reaction under the alkaline electrolyte (0.1 M KOH). Although an a-KB exhibited high selectivity to H_2O_2 from ORR in the alkaline condition, no products from CH_4 oxidation were observed. This implies that the coupling between ORR and CH_4 oxidation reaction is influenced by the pH of the electrolyte, favoring the acidic condition for the EMPO reaction (Supplementary Fig. 17-18).

[page 22 line 3-5] Cyclic voltammetry (CV) was performed between 0.05 and 1.20 V (vs. RHE) in N_2 -saturated 0.05 M H_2SO_4 and 0.1 M KOH at a scan rate of 100 mV s^{-1} for 10 cycles, in which steady CV response was obtained.

[page 22 line 7-10] The H_2O_2 production activity was assessed by linear sweep voltammetry (LSV) from 1.1 to 0.2 V (vs. RHE) in O_2 -saturated 0.05 M H_2SO_4 and 0.1 M KOH at a scan rate of 5 mV s^{-1} and rotating speed of 1600 rpm.

[page 24 line 18-19] For the alkaline EMPO experiment, all reaction conditions were the same except for applying 0.1 M KOH as an electrolyte instead of 0.05 M H_2SO_4 .

5. (minor) Fig. 2c discussed after Figs 2d-f. Please rearrange figures order.

Response: Thank you for your careful comments. Now, we rearranged the figure order. We moved Fig. 2c to Fig. 2f. The main manuscript is revised in the order of Fig.2.

6. Reaction mechanism discussed/elaborated in lines 293-243 needs to be revised according to the comments mentioned above.

Response: Thanks to the reviewer's comments, we were able to improve understanding of the reaction mechanism. Based on the new experimental results and reported articles, the proposed EMPO reaction mechanism was modified. The revised manuscript is already mentioned in response #3. Thank you for your comment.

7. The results shown under the "Electro-assisted C₂H₆ partial oxidation" section are difficult to be related to CH₄. C-H activation of CH₄ is significantly more challenging than that of C₂H₆. The authors need to provide more convincing reasoning for including C₂H₆ oxidation in this study.

Response: Thank you for your comments that more convincing reasoning for including C₂H₆ oxidation experiment in this study is needed. If we demonstrate electro-assisted C₂H₆ partial oxidation and propose the possibility of its application for CH₄ partial oxidation, then it might not be convincing as the reviewer mentioned. Here, however, we demonstrate C-H activation for both CH₄ and C₂H₆, respectively. We include electro-assisted C₂H₆ partial oxidation reaction in this study for the following two purposes; i) identification of carbon source from C-H bond cleavage, and ii) demonstration of the general tendency of partial oxidation of alkane. To identify carbon source, we performed the labelled ¹³CH₄ experiment which provides direct evidence but is expensive. Partial oxidation products of C₂H₆ include C₂H₅OH, C₂H₅OOH, and CH₃COOH. The exclusive observation of the C₂ chemical products such as C₂H₅OH and C₂H₅OOH are supporting direct C-H activation from C₂H₆. The reaction products changed from C₁ to C₂ simply by replacing feed gas from CH₄ to C₂H₆ under the same reaction condition. Although there is a possibility that different activity between CH₄ and C₂H₆ activation reaction because C-H bond dissociation energy (439.3 kJ mol⁻¹) of CH₄ is larger than that of C₂H₆ (421.7 kJ mol⁻¹), we propose electro-assisted partial oxidation of C₂H₆ can be utilized to test the feasibility of C-H bond partial oxidation. At the same time, the products of C₂H₆ partial

oxidation are similar to those of CH₄ (CH₃OH, CH₃OOH, and HCOOH), which suggests the electro-assisted partial oxidation tendency between CH₄ and C₂H₆ is similar and applicable to C₂H₆ upgrading. “Electro-assisted C₂H₆ partial oxidation” experiment might not be out of scope, and we could give useful insights into the general tendency of electro-assisted alkane partial oxidation.

To clarify these points, we added the discussion in the revised manuscript (page 17 line 6-8, page 17 line 11-15). Thank you for your comment.

[page 17 line 6-8] To validate the electro-assisted C–H activation of alkane for partial oxidation products generation, the C₂H₆ partial oxidation was examined under the same cathodic H₂O₂ generation conditions.

[page 17 line 11-15] The exclusive observation of the C₂ partial oxidation products are supporting direct C–H activation from C₂H₆. The reaction products were changed from C₁ (CH₃OH, CH₃OOH, and HCOOH) to C₂ (C₂H₅OH, C₂H₅OOH, and CH₃COOH) with similar oxidation states by simply replacing CH₄(g) with C₂H₆(g) under the same reaction conditions. The reaction pathway is similar to that of EMPO.

8. Cycle stability experiment is understandable for photo(electro)-catalysis. However, ORR experiments (and electrocatalysis studies in general) are usually presented in one run over long operation time to comment on the catalyst as well as overall system stability.

Response: As the reviewer mentioned, electrocatalysis usually shows catalytic stability by one run over a long operation time. To verify the stability of the EMPO system, we performed the EMPO reaction for six hours, and electrolyte-flowing H-cell setup was applied (as shown in Figure R7). During the reaction, no decreases in the current density as well as production rate of HCOOH were observed showing stable operation of the EMPO system (Figure R8).

Figure R7. The electrolyte-flowing H-cell setup for the stability operation experiment.

Figure R8. Stability test of the EMPO system. Current density, and product amount and production rate of EMPO reaction during 6 hours. Reaction conditions: 25 °C, 1 bar, O₂: 100 sccm, CH₄: 100 sccm, 550 mL of 0.05 M H₂SO₄ circulated by 10 mL min⁻¹, stirred at 700 rpm, and 0 V (vs. RHE).

We revised our manuscript to include the details of the results and the experimental method (page 18 line 12-16, page 20 line 14-15, Fig.5, and Supplementary Fig. 20-21). Thank you for your comment.

[page 18 line 12-16] A stability test of the EMPO system was performed (Fig. 5). The stability test was investigated under the electrolyte-flowing H-cell setup to keep HCOOH concentration low because high concentration of HCOOH induces over-oxidation to CO₂ (Supplementary Fig. 20-21). During six hours of the EMPO reaction, no decreases in the current density or production rate of HCOOH were observed demonstrating its stable operation.

[page 20 line 14-15] This EMPO system applying metal-free carbon catalyst showed significant stable performance for 6 hours.

9. Were there any gaseous products from the EMPO? CO₂ is expected to be produced from inevitable over-oxidation of CH₄.

Response: As the reviewer mentioned, CO₂ is expected to be produced due to the over-oxidation of CH₄. Therefore, we analyzed gas products by an on-line connected gas chromatography (GC). To qualify the gas products, the outflow gas from the H-cell of the EMPO reaction was sampled to inject the inlet of GC. A flame ionization detector (FID) after a methanizer was used to increase CO detection sensitivity. The chromatogram (Figure R9) shows the peak corresponding to CO₂ appeared during EMPO reaction, confirming that CO₂ is produced from over-oxidation of CH₄.

Figure R9. Gas chromatogram of gas products during EMPO.

Then, we measured the amount of CO₂ depending on the reaction time. To quantify the CO₂, the GC was calibrated using standard gases (10~100 ppm CO and CO₂ gases) (Figure R10).

Figure R10. Calibration curves of the GC for CO and CO₂ gases.

The concentration of CO₂ in outflow gas was significantly lower compared to HCOOH, and it was slightly increased as that of HCOOH was increased (Figure R11). Then CO₂ concentration was kept constant once the concentration of HCOOH reached saturated value confirming that increased concentration of HCOOH induces over-oxidation to CO₂. The total mole of produced CO₂ gas (n_{CO_2}) during reaction time can be calculated as follows:

$$n_{CO_2} = \int C_{CO_2} \cdot V dt$$

where C_{CO_2} , V , and t are CO₂ gas concentration, gas flow rate (200 sccm), and time, respectively.

Figure R11. Gas (CO₂) and liquid (HCOOH) concentration changes upon reaction time.

The production rate of CO₂ was 0.76 μmol h⁻¹ and the selectivity of CO₂ was 4.4% (Figure R12).

Figure R12. Production rate of total products including CO₂.

Above findings are added to the manuscript (page 10 line 17-21, page 18 line 12-15, page 26 line 8-14, and Supplementary Fig. 6-7, 20, 24). Thank you for your comment.

[page 10 line 17-21] Additionally, we investigated whether gas products such as CO and CO₂ were generated due to the over-oxidation of CH₄ by on-line gas chromatography (GC). During the EMPO reaction, the amount of the produced CO₂ was 0.76 µmol h⁻¹ which was an insignificant amount compared to that of HCOOH (13.8 µmol h⁻¹) (Supplementary Fig. 6-7).

[page 18 line 12-15] The stability test was investigated under the electrolyte-flowing H-cell setup to keep HCOOH concentration low because high concentration of HCOOH induces over-oxidation to CO₂ (Supplementary Fig. 20-21).

[page 26 line 8-14] GC analysis of gas products

The gas products from the EMPO reaction were quantified by on-line gas chromatography (GC, Agilent 6890). A flame ionization detector (FID) was used to detect CO and CO₂. A methanizer was utilized to increase the detection sensitivity of CO and CO₂. The GC system was equipped with a ShinCarbon ST column (Restek) to separate gas products. The calibration of the GC was carried out by flowing three calibration gas mixtures with CO- and CO₂-concentrations ranging from 10 to 100 ppm (Supplementary Fig. 24).

[Supplementary Figure 20] The concentration of CO₂ was significantly lower compared to HCOOH, and it was slightly increased as that of HCOOH was increased. Then CO₂ concentration was kept constant once the concentration of HCOOH reached saturated value confirming that increased concentration of HCOOH induces over-oxidation to CO₂.

10. How was the 83.6% selectivity of HCOOH calculated? And does selectivity here includes all products or limited to liquid products (liquid product selectivity).

Response: The reviewer asked how to calculate the selectivity value of HCOOH from the EMPO reaction. For the activity analysis of our EMPO reaction, we measured the production rate ($18.9 \mu\text{mol h}^{-1}$) of the oxygenate products and selectivity (83.6%) of HCOOH among the liquid products (CH₃OH, CH₃OOH, HCOOH). In the original manuscript, the selectivity was calculated based on measured liquid products (limited to liquid products) according to the below equation:

$$\text{Selectivity (\%)} = \frac{\text{Mole of HCOOH}}{\text{Total mole of liquid products (CH}_3\text{OH, CH}_3\text{OOH, HCOOH)}}$$

Now we calculate the selectivity including gas products according to the below equation:

$$\text{Selectivity (\%)} = \frac{\text{Mole of HCOOH}}{\text{Total mole of all products (CH}_3\text{OH, CH}_3\text{OOH, HCOOH, CO}_2\text{)}}$$

The selectivity of HCOOH becomes 80.7% at 0V (vs. RHE), and the manuscript is revised (page 7 line 4-6 and page 20 line 12-14). Thank you for your comment.

[page 7 line 4-6] *In the EMPO system, most of the produced CH₃OOH is electrochemically reduced to CH₃OH on the cathode, and CH₃OH is further oxidized to HCOOH, resulting in a high selectivity to HCOOH (80.7%).*

[page 20 line 12-14] *Additionally, the unstable CH₃OOH product can be converted to CH₃OH on the cathode at the EMPO working potential; this significantly improved the product selectivity (80.7%) toward the stable liquid fuel, HCOOH.*

11. (minor) in line 27 in the abstract “Here, we present a novel approach...”. The approach is not novel as mentioned in lines 90-94. So, please rewrite this sentence.

Response: As the reviewer suggested, we delete “novel” in the sentence as below (page 3 line 5-8). Thank you for your comment.

[page 3 line 5-8] *Here, we propose an electro-assisted approach for the partial oxidation of methane, viz., electro-assisted methane partial oxidation (EMPO) using in-situ cathodically generated H₂O₂, which can be accomplished at ambient pressure and temperature.*

Reviewer #2: In the manuscript, Jong Hyeok Park et al reported an approach for the partial oxidation of methane, viz., electro-assisted methane partial oxidation (EMPO) using in-situ cathodically generated H_2O_2 , which can be accomplished at ambient pressure and temperature. Upon using activated carbon as the electrocatalyst, the EMPO process enables the partial oxidation of methane in an acidic electrolyte to produce oxygenated liquid products. However, the mechanism and some characterizations of this paper are unclear enough. The authors should consider the following specific comments in a possible revision and submit the revised version to more specialized journals.

1. The H_2O_2 produced in situ does not have ability to activate the methane C-H bond, so it needs to be activated to produce active species, such as $\cdot\text{OH}$. However, this paper did not define the active species, and the explanation of H_2O_2 activation mechanism and methane activation mechanism was insufficient.

Response: Thank you for your valuable comments to consider active species, and activation mechanisms of H_2O_2 and CH_4 . We apologize for our insufficient description, and we also agree with the reviewer's points that active species which have the ability to activate the C-H bond of CH_4 should be defined. In order to investigate whether active species such as $\cdot\text{OH}$ radicals can be formed during electrochemical oxygen reduction reaction (ORR), we performed more analyses and control experiments to identify active species which can activate methane C-H bond.

First, trapping experiments were performed by adding tert-butyl alcohol (TBA) and 1,4-benzoquinone (BQ) as $\cdot\text{OH}$ and $\cdot\text{OOH}$ radical scavengers in the electrolytes, respectively, to determine whether reactive oxygen species (ROS) can be formed and it can affect the activity of EMPO reaction. If the EMPO reaction includes a radical process, the amount of reaction products should be affected because radicals are trapped by radical scavengers. Because CH_3OOH , which contains a methyl group (CH_3-), can be formed directly through the partial oxidation of CH_4 , we measured the product amount of CH_3OOH in the presence or absence of the radical scavengers. As you can see in Figure R1, no CH_3OOH was detected under an excess amount (400 μmol) of scavengers and the product amount of CH_3OOH was decreased in the presence of both TBA and BQ (40 μmol), respectively, supporting that CH_4 activation and CH_3OOH formation include radical process.

Figure R1. CH_3OOH production after EMPO reaction in the presence of a) 400 μmol and b) 40 μmol of scavengers. TBA and BQ are radical scavengers trapping $\cdot\text{OH}$ and $\cdot\text{OOH}$, respectively.

Next, to verify the radical species in a more direct manner, we additionally conducted electron paramagnetic resonance (EPR) analysis by adding 5,5-dimethyl-1-pyrroline N-oxide (DMPO) in the electrolytes as a spin trap. EPR signals were collected by measuring electrolytes before and after applying reduction potential (0 V vs. RHE) flowing O_2 and O_2+CH_4 with adding DMPO (Figure R2). While no radical species was observed in the electrolyte before the reaction (grey line), a quartet spectrum with a relative intensity ratio of 1:2:2:1 appeared after applying potential under O_2+CH_4 flowing (blue line). It is typically attributed to the formation of DMPO–OH adducts supporting that radicals were formed during EMPO reaction (*Free Radic. Biol. Med.* **3**, 259-303 (1987)). Especially, DMPO–OH adducts were also detected when only O_2 was fed except CH_4 (red line). This supports that $\cdot\text{OH}$ radicals were generated in our ORR condition, and the previous study also reported electrochemical formation of the active species during ORR (*Angew. Chem. Int. Ed.* **60**, 10375-10383 (2021) / *J. Phys. Chem. Lett.* **12**, 7797-7803 (2021)). Although the signals corresponding to DMPO–OH adducts was the only EPR signal detected, the formation of other radical species cannot be excluded because other DMPO adducts have a short lifetime and quickly decompose to the DMPO–OH adduct (*Free Radic. Res. Commun.* **19**, S79-S87 (1993)).

Figure R2. EPR spectra of the electrolytes before (grey line) and after applying potential (0 V vs. RHE) under O₂(g) (red line) and O₂(g)+CH₄(g) (blue line) flowing with DMPO.

From these above experiments and reported literatures (*Angew. Chem. Int. Ed.* **60**, 10375-10383 (2021) / *ACS Catal.* **8**, 7961-7972 (2018)), we could define the active species, ·OH and ·OOH, and propose the activation mechanism of H₂O₂ and CH₄ as below (Figure R3):

H₂O₂ activation:

CH₄ activation:

Figure R3. Proposed mechanism. Schematic illustration of the reaction mechanism for selective HCOOH production by EMPO.

We now modify the manuscript to discuss activation pathways during the EMPO reaction (page 3 line 13-15, page 7 line 1-2, page 12 line 19-page 13 line 5, page 13 line 9-page 14 line 5, page 16 line 5-13, page 20 line 7-8, page 20 line 10-12, page 24 line 7-13, page 26 line 15-19, Fig. 2b-c, 3, and Supplementary Fig. 12). Thank you for your comment.

[page 3 line 13-15] The mechanistic study revealed that reactive oxygen species (ROSs) such as $\cdot OH$ and $\cdot OOH$ radicals are produced during the reaction and activated methane and methanol.

[page 7 line 1-2] Reactive oxygen species (ROSs) (i.e., $\cdot OH$ and $\cdot OOH$ radicals) were generated during ORR and activated CH_4 and CH_3OH .

[page 12 line 19-page 13 line 5] In contrast with EMPO, no liquid products were formed when $c\text{-}H_2O_2$ was added in the reaction cell at 25 °C (Fig. 2a and Supplementary Fig. 9-10) implying that H_2O_2 molecule itself cannot react with CH_4 and ROSs formed during electrochemical ORR are crucial for the EMPO.^{37, 38}

We performed control experiments to figure out whether ROSs are generated from O_2 and participate in the EMPO reaction.

[page 13 line 9-page 14 line 5] Then, to verify radical species more directly, we additionally

conducted electron paramagnetic resonance (EPR) analysis by adding 5,5-dimethyl-1-pyrroline N-oxide (DMPO) in the electrolytes as a spin trap. EPR signals were collected by measuring electrolytes before and after applying reduction potential (0 V vs. RHE) flowing $O_2(g)$ or $O_2(g)+CH_4(g)$ under the addition of DMPO (Fig. 2b). We confirmed that no radical signal was observed in the electrolyte before the reaction (grey line). Meanwhile, a quartet spectrum with a relative intensity ratio of 1:2:2:1 appeared after applying potential under O_2+CH_4 feeding (blue line). It is typically attributed to the formation of DMPO–OH adducts supporting that radicals were formed during EMPO reaction.³⁹ Especially, DMPO–OH adducts were also detected when only O_2 was fed without CH_4 suggesting that ROSs are generated as the result of electrochemical ORR in our system (red line). Although the signal corresponding to DMPO–OH adducts was the only EPR signal detected, the formation of other ROSs cannot be excluded because other DMPO–ROS adducts have a short lifetime and quickly decompose to the DMPO–OH adduct.⁴⁰ In addition, we further performed trapping experiments applying tert-butyl alcohol (TBA) and 1,4-benzoquinone (BQ) as $\cdot OH$ and $\cdot OOH$ radical scavengers, respectively. To identify the effects of the ROSs, we measured the product amount of CH_3OOH in the presence or absence of the radical scavengers. No CH_3OOH was detected under an excess amount (400 μmol) of scavengers (Fig. 2c). When 40 μmol of TBA and BQ were present in the electrolytes, respectively, the generated amount of CH_3OOH was decreased significantly compared to the case without these scavengers (Supplementary Fig. 12). These series results indicate that ROSs generated from O_2 activate CH_4 to produce CH_3OOH .

[page 16 line 5-13] As illustrated in Fig. 3, first, O_2 is reduced to H_2O_2 on the electrode through the electrochemical two-electron pathway ORR, and ROSs are formed (ROSs formation). Second, $\cdot OH$ radicals activate CH_4 to produce $CH_3\cdot$ radicals in the electrolyte (CH_4 activation). Then, CH_3OOH is generated by the reaction between $CH_3\cdot$ and $\cdot OOH$ radicals (CH_3OOH formation). Subsequently, CH_3OH was generated by electrochemical reduction of CH_3OOH and radical reaction between $CH_3\cdot$ and $\cdot OH$. Finally, $HCOOH$ was formed in the presence of $\cdot OH$ radicals ($HCOOH$ formation). Thus, $HCOOH$ can be generated as a selective reaction product with the aid of electrochemical reduction potential at room temperature and atmospheric pressure in the EMPO system.

[page 20 line 7-8] CH_4 can be activated by the in-situ cathodically generated ROSs (i.e., $\cdot OH$ and $\cdot OOH$ radicals) from the O_2 reduction.

[page 20 line 10-12] Through a mechanistic study of the EMPO process, we propose that ROSs

are the active species of the partial oxidation of CH₄.

[page 24 line 7-13] For the EPR experiment, 1mmol of DMPO (Sigma-Aldrich) was applied as a spin trap before the reaction. Other reaction conditions were the same except for feeding O₂ or O₂+CH₄ gases.

For the trapping experiments, 400 and 40 μmol of TBA (Sigma-Aldrich, ≥99.5%) and BQ (Sigma-Aldrich, ≥98%) were added before the reaction as ·OH and ·OOH radical scavengers, respectively. Other reaction conditions were the same.

[page 26, line 15-19] **EPR experiment**

The EPR spectrum of electrolytes was measured at KBSI Seoul Western Center using CW/Pulse EPR system with the following parameters: frequency 9.852 GHz; power 3 mW; modulation amplitude 1 G; time constant 20.48 ms; conversion time 20.00 ms; scan 8; temperature RT.

2. Supplementary Figure 7 is not described in the text. Please complete it.

Response: Thank you for your careful comments. We moved Supplementary Figure 7 to Supplementary Figure 25 and now include the description (page 27 line 5-6).

[page 27 line 5-6] The synthesized CH₃OOH was confirmed and quantified by ¹H-NMR analysis (Supplementary Fig. 25).

3. There is a mixture of upper and lower case in the title of references. Please carefully check the format of the reference document and modify it.

Response: Thank you for your comment. We modified a mixture of upper and lower case in the title of references to fit the format of the reference document.

4. Some tenses are inaccurate, should use simple past tense, please carefully check the tenses of the whole text and carefully modify.

Response: Thank you for your careful comments on the manuscript, and we revised the grammar error of the revised manuscript (page 14 line 9-11, page 23 line 19-20, page 24 line 15, page 24 line 16-17, page 24 line 20-21, and page 25 line 4-6).

[page 14 line 9-11] Based on these results, we speculated that the EMPO system not only guides the reaction between CH_4 and electrochemically generated H_2O_2 , but also alters the selectivity of the reaction and enhances the production rate.

[page 23 line 19-20] For the $^{13}\text{CH}_4$ isotope experiment, all reaction conditions were the same except for feeding $^{13}\text{CH}_4$ instead of $^{12}\text{CH}_4$.

[page 24 line 15] Other reaction conditions were the same.

[page 24 line 16-17] For the CH_3OH oxidation experiment, all reaction conditions were the same except for injecting 100 μmol of CH_3OH (Sigma-Aldrich, $\geq 99.9\%$) before the reaction instead of CH_4 .

[page 24 line 20-21] For the electro-assisted C_2H_6 partial oxidation experiment, all reaction conditions were the same except feeding C_2H_6 instead of CH_4 .

[page 25 line 4-6] Ar was purged instead of O_2 to keep balance of partial flow of CH_4 . Commercial H_2O_2 (30 wt% in H_2O , Sigma-Aldrich) was injected during reaction time (30 min) via syringe pump.

Reviewer #3: In the submission the authors proposed an electro-assisted methane partial oxidation (EMPO) approach for the partial oxidation at ambient pressure and temperature using in-situ cathodically generated H_2O_2 . Using activated carbon as the electrocatalyst, the EMPO process enabled the partial oxidation of methane in an acidic electrolyte to selectively produce formic acid (83.6%). A reaction mechanism with unstable methyl peroxide generated from methane partial oxidation as the intermediate was proposed. Such an electro-assisted oxidation process was also demonstrated effective for ethane partial oxidation.

The reaction data are interesting and the results are presented in a clear way, but the mechanistic studies are not sufficient. The following issues must be clarified before the submission can be accepted:

1. Were CO and CO_2 produced during the catalytic reaction? How about the carbon balance?

Response: Thank you for the comments on the carbon balance. During the revision, we analyzed the production of the gas products to examine whether CO and CO_2 are produced due to the over-oxidation of CH_4 . We performed gas products analysis by a gas chromatography (GC) which is on-line connected with the electrochemical cell of the EMPO reaction. To qualify the gas products, we measured the outflow gas of the EMPO reaction by a flame ionization detector (FID) with a methanizer to increase the sensitivity for CO or CO_2 . The peak corresponding to CO showed no change during the EMPO reaction. However, we observed the production of CO_2 gas due to the over-oxidation of CH_4 (Figure R1).

Figure R1. Gas chromatography of gas products during EMPO.

To quantify the CO_2 , the GC was calibrated using standard gases (10~100 ppm CO and CO_2

gases) (Figure R2).

Figure R2. Calibration curves of the GC for CO and CO₂ gas

Then, we measured the concentration of CO₂ depending on reaction time, showing 0.127 $\mu\text{mol L}^{-1}$ and 0.223 $\mu\text{mol L}^{-1}$ after 30 min and 1hour of the EMPO reaction, respectively. The concentration of CO₂ in outflow gas was slightly increased as that of HCOOH was increased (Figure R3). CO₂ concentration was then kept constant once the concentration of HCOOH reached saturated value, supporting that the increased concentration of HCOOH causes over-oxidation to CO₂. The total mole of produced CO₂ gas (n_{CO_2}) during reaction time can be calculated as follows:

$$n_{CO_2} = \int C_{CO_2} \cdot V dt$$

where C_{CO_2} , V , and t are CO₂ gas concentration, gas flow rate (200 sccm), and time, respectively.

Figure R3. Gas (CO₂) and liquid (HCOOH) concentration changes upon reaction time.

The production rate of CO₂ was 0.76 μmol h⁻¹, and the selectivity of CO₂ was 4.4% (Figure R4). The produced amount of CO₂ was insignificant compared to that of HCOOH (13.79 μmol h⁻¹).

Figure R4. Production rate of total products including CO₂.

As the reviewer asked the carbon balance, we calculate the HCOOH product selectivity over all products (*CH₃OH*, *CH₃OOH*, *HCOOH*, *CO₂*). Because all product molecules are C1 chemical, the selectivity of HCOOH based on all products was calculated by the equation below:

$$\text{Selectivity (\%)} = \frac{\text{Mole of HCOOH}}{\text{Total mole of all products (CH}_3\text{OH, CH}_3\text{OOH, HCOOH, CO}_2\text{)}}$$

The selectivity of HCOOH is revised to 80.7% at 0V (vs. RHE).

Now, we include the measurement of CO₂ gas and product selectivity in the revised manuscript. Above findings are added to the manuscript (page 10 line 17-21, page 18 line 12-15, page 26 line 8-14 and Supplementary Fig. 6-7, 20, and 24). Thank you for your comment.

[page 10 line 17-21] Additionally, we investigated whether gas products such as CO and CO₂ were generated due to the over-oxidation of CH₄ by on-line gas chromatography (GC). During the EMPO reaction, the amount of the produced CO₂ was 0.76 μmol h⁻¹ which was an insignificant amount compared to that of HCOOH (13.8 μmol h⁻¹) (Supplementary Fig. 6-7).

[page 18 line 12-15] The stability test was investigated under the electrolyte-flowing H-cell setup to keep HCOOH concentration low because high concentration of HCOOH induces over-oxidation to CO₂ (Supplementary Fig. 20-21).

[page 26 line 8-14] GC analysis of gas products

The gas products from the EMPO reaction were quantified by on-line gas chromatography (GC, Agilent 6890). A flame ionization detector (FID) was used to detect CO and CO₂. A methanizer was utilized to increase the detection sensitivity of CO and CO₂. The GC system was equipped with a ShinCarbon ST column (Restek) to separate gas products. The calibration of the GC was carried out by flowing three calibration gas mixtures with CO- and CO₂-concentrations ranging from 10 to 100 ppm (Supplementary Fig. 24).

[Supplementary Figure 20] The concentration of CO₂ was significantly lower compared to HCOOH, and it was slightly increased as that of HCOOH was increased. Then CO₂ concentration was kept constant once the concentration of HCOOH reached saturated value confirming that increased concentration of HCOOH induces over-oxidation to CO₂.

2. The authors used activated carbon as the electrocatalyst. Although ¹³CH₄ was used to demonstrated that the products came from CH₄, it could not be excluded that the carbon atoms in the catalyst participated into the reaction because the chemical bonds in the activated carbon are weaker than that in CH₄. Is it possible to use none carbon-contained catalysts to test the idea?

Response: Thank you for suggesting to test non-carbon-contained catalysts to clarify the carbon source. As the reviewer mentioned, to exclude the possibility that the carbon atom in the catalyst participates in the EMPO reaction, we tested an Au foil as a carbon-free catalyst. As you can see in Figure R5, HCOOH was generated during the EMPO reaction by applying Au foil for ORR catalyst confirming that HCOOH originated from CH₄ oxidation. We also presented Ar-experiment instead of CH₄ and ¹³CH₄ experiment in the original manuscript, to verify the reaction products originated from CH₄. All these experiments consistently support that HCOOH is produced from CH₄ partial oxidation.

Figure R5. ^1H -NMR analysis after EMPO reaction applying Au foil as a carbon-free ORR catalyst. The peaks at 0.17, 2.71, and 8.22 ppm are attributed to CH_4 , $(\text{CH}_3)_2\text{SO}$ (DMSO), and HCOOH , respectively.

Above finding is added to the manuscript (page 10 line 9-12, page 23 line 21-22, and Supplementary Fig. 5). Thank you for your comment.

[page 10 line 9-12] We conducted an EMPO reaction applying Au foil as a carbon-free ORR catalyst, and the HCOOH was also generated as a liquid product confirming that the reaction products are generated from CH_4 oxidation (Supplementary Fig. 5).

[page 23 line 21-22] For the carbon-free ORR catalyst experiment, all reaction conditions were the same except for applying Au foil (Dasom RMS, 99.99%) instead of a-KB catalyst.

3. The authors proposed in-situ cathodically generated H_2O_2 as the reactive species, however, is it possible that the more reactive OOH species, a likely intermediate during H_2O_2 production, directly acted as the active species?

Response: Thank you for your valuable comments to consider the presence of reactive oxygen species (ROSs) during EMPO. Since ROSs such as $\cdot\text{OH}$ and $\cdot\text{OOH}$ radicals can be formed during electrochemical oxygen reduction reaction (ORR) (*Angew. Chem. Int. Ed.* **60**, 10375-10383 (2021) / *J. Phys. Chem. Lett.* **12**, 7797-7803 (2021)), we designed control experiments to figure out whether ROSs can be formed during the EMPO reaction. First, we conducted trapping experiments applying tert-butyl alcohol (TBA) and 1,4-benzoquinone (BQ) as $\cdot\text{OH}$ and $\cdot\text{OOH}$ radical scavengers, respectively. If the ROSs are generated and directly act as the

active species, the amount of reaction products should be affected because ROSs are trapped by radical scavengers. Because CH_3OOH , which contains a methyl group (CH_3-), can be formed directly through the partial oxidation of CH_4 , we measured the product amount of CH_3OOH in the presence or absence of the radical scavengers. As you can see in Figure R6, no CH_3OOH was detected under excess amount (400 μmol) of scavengers and product amount of CH_3OOH was decreased in the presence of both TBA and BQ (40 μmol), respectively, supporting that CH_4 activation and CH_3OOH formation include radical process.

Figure R6. CH_3OOH production after EMPO reaction in the presence of a) 400 μmol and b) 40 μmol of scavengers. TBA and BQ are radical scavengers which can trap $\cdot\text{OH}$ and $\cdot\text{OOH}$, respectively.

Then, to verify radical species directly, we additionally conducted electron paramagnetic resonance (EPR) analysis by adding 5,5-dimethyl-1-pyrroline N-oxide (DMPO) in the electrolytes as a spin trap. EPR signals were collected by measuring electrolytes before and after applying reduction potential (0 V vs. RHE) flowing O_2 and O_2+CH_4 , respectively, with adding DMPO (Figure R7). While no radical was observed in the electrolyte before the reaction (grey line), a quartet spectrum with a relative intensity ratio of 1:2:2:1 appeared after applying potential under O_2+CH_4 flowing (blue line). It is typically attributed to the formation of DMPO–OH adducts (*Free Radic. Biol. Med.* **3**, 259-303 (1987)) confirming that ROSs were formed during EMPO reaction. Especially, DMPO–OH adducts were also detected when only O_2 was fed except CH_4 showing that ROSs are generated during electrochemical ORR reaction (red line). Although the signals corresponding to DMPO–OH adducts was the only EPR signal detected, the formation of other ROS cannot be excluded because other DMPO–ROS adducts

have a short lifetime and quickly decompose to the DMPO–OH adduct (*Free Radic. Res. Commun.* **19**, S79-S87 (1993)).

Figure R7. EPR spectra of the electrolytes before (grey) and after applying potential under O_2 (red) and O_2+CH_4 (blue) flowing with DMPO.

From these above experiments, we propose that ROSs are formed during EMPO reaction and they activate CH_4 generating CH_3OOH (*ACS Catal.* **8**, 7961-7972 (2018)). Additionally, given that reported literatures (*Nat. Catal.* **1**, 889-896 (2018) / *Angew. Chem. Int. Ed.* **52**, 1280-1284 (2013) / *ACS Catal.* **11**, 6684-6691 (2021)), CH_3OH and $HCOOH$ also can be generated via radical processes.

The modified proposed EMPO reaction mechanism includes ROSs for $HCOOH$ production as follows:

ROSs formation:

CH_4 activation:

CH_3OOH formation:

CH₃OH formation:

HCOOH formation:

The schematic illustration of the revised mechanism is shown in Figure R8.

Figure R8. Proposed mechanism. Schematic illustration of the reaction mechanism for selective HCOOH production by EMPO.

Above findings are added to the manuscript (page 3 line 13-15, page 7 line 1-2, page 12 line 19-page 13 line 5, page 13 line 9-page 14 line 5, page 16 line 5-13, page 20 line 7-8, page 20 line 10-12, page 24 line 7-13, page 26 line 15-19, Fig. 2b-c, 3 and Supplementary Fig. 12) as below. Thank you for your comment.

[page 3 line 13-15] The mechanistic study revealed that reactive oxygen species (ROSs) such as $\cdot\text{OH}$ and $\cdot\text{OOH}$ radicals are produced during the reaction and activated methane and

methanol.

[page 7 line 1-2] Reactive oxygen species (ROSs) (i.e., $\cdot\text{OH}$ and $\cdot\text{OOH}$ radicals) were generated during ORR and activated CH_4 and CH_3OH .

[page 12 line 19-page 13 line 5] In contrast with EMPO, no liquid products were formed when $c\text{-H}_2\text{O}_2$ was added in the reaction cell at $25\text{ }^\circ\text{C}$ (Fig. 2a and Supplementary Fig. 9-10) implying that H_2O_2 molecule itself cannot react with CH_4 and ROSs formed during electrochemical ORR are crucial for the EMPO.^{37, 38}

We performed control experiments to figure out whether ROSs are generated from O_2 and participate in the EMPO reaction.

[page 13 line 9-page 14 line 5] Then, to verify radical species more directly, we additionally conducted electron paramagnetic resonance (EPR) analysis by adding 5,5-dimethyl-1-pyrroline N-oxide (DMPO) in the electrolytes as a spin trap. EPR signals were collected by measuring electrolytes before and after applying reduction potential (0 V vs. RHE) flowing $\text{O}_2(\text{g})$ or $\text{O}_2(\text{g})+\text{CH}_4(\text{g})$ under the addition of DMPO (Fig. 2b). We confirmed that no radical signal was observed in the electrolyte before the reaction (grey line). Meanwhile, a quartet spectrum with a relative intensity ratio of 1:2:2:1 appeared after applying potential under O_2+CH_4 feeding (blue line). It is typically attributed to the formation of DMPO–OH adducts supporting that radicals were formed during EMPO reaction.³⁹ Especially, DMPO–OH adducts were also detected when only O_2 was fed without CH_4 suggesting that ROSs are generated as the result of electrochemical ORR in our system (red line). Although the signal corresponding to DMPO–OH adducts was the only EPR signal detected, the formation of other ROSs cannot be excluded because other DMPO–ROS adducts have a short lifetime and quickly decompose to the DMPO–OH adduct.⁴⁰ In addition, we further performed trapping experiments applying tert-butyl alcohol (TBA) and 1,4-benzoquinone (BQ) as $\cdot\text{OH}$ and $\cdot\text{OOH}$ radical scavengers, respectively. To identify the effects of the ROSs, we measured the product amount of CH_3OOH in the presence or absence of the radical scavengers. No CH_3OOH was detected under an excess amount (400 μmol) of scavengers (Fig. 2c). When 40 μmol of TBA and BQ were present in the electrolytes, respectively, the generated amount of CH_3OOH was decreased significantly compared to the case without these scavengers (Supplementary Fig. 12). These series results indicate that ROSs generated from O_2 activate CH_4 to produce CH_3OOH .

[page 16 line 5-13] As illustrated in Fig. 3, first, O_2 is reduced to H_2O_2 on the electrode through the electrochemical two-electron pathway ORR, and ROSs are formed (ROSs formation).

Second, ·OH radicals activate CH₄ to produce CH₃· radicals in the electrolyte (CH₄ activation). Then, CH₃OOH is generated by the reaction between CH₃· and ·OOH radicals (CH₃OOH formation). Subsequently, CH₃OH was generated by electrochemical reduction of CH₃OOH and radical reaction between CH₃· and ·OH. Finally, HCOOH was formed in the presence of ·OH radicals (HCOOH formation). Thus, HCOOH can be generated as a selective reaction product with the aid of electrochemical reduction potential at room temperature and atmospheric pressure in the EMPO system.

[page 20 line 7-8] CH₄ can be activated by the in-situ cathodically generated ROSs (i.e., ·OH and ·OOH radicals) from the O₂ reduction.

[page 20 line 10-12] Through a mechanistic study of the EMPO process, we propose that ROSs are the active species of the partial oxidation of CH₄.

[page 24 line 7-13] For the EPR experiment, 1mmol of DMPO (Sigma-Aldrich) was applied as a spin trap before the reaction. Other reaction conditions were the same except for feeding O₂ or O₂+CH₄ gases.

For the trapping experiments, 400 and 40 μmol of TBA (Sigma-Aldrich, ≥99.5%) and BQ (Sigma-Aldrich, ≥98%) were added before the reaction as ·OH and ·OOH radical scavengers, respectively. Other reaction conditions were the same.

*[page 26, line 15-19] **EPR experiment***

The EPR spectrum of electrolytes was measured at KBSI Seoul Western Center using CW/Pulse EPR system with the following parameters: frequency 9.852 GHz; power 3 mW; modulation amplitude 1 G; time constant 20.48 ms; conversion time 20.00 ms; scan 8; temperature RT.

4. In the controlled c-H₂O₂ system experiments, c-H₂O₂ was supplied with a syringe pump and injected at the same rate as the H₂O₂ production rate in the e-H₂O₂ system, and no liquid products were generated. The authors proposed that the stabilizers contained in c-H₂O₂ (stannate- and phosphorus-containing compounds) inhibited the decomposition of H₂O₂. Where did the proposed stabilizers come from? And what happened if the H₂O₂ supply was increased?

Response: Thank you for the comments on the commercial H₂O₂ experiments. H₂O₂ can decompose to H₂O and O₂ in a catalyzed reaction, of which the active catalysts are trace amounts of transition metal ions, such as copper, iron, and cobalt ions. The stabilizers such as stannate- and phosphorus-containing compounds (i.e., sodium stannate, phosphoric acids and

their salts) are normally added to commercial hydrogen peroxide solutions. They form metal complexes with metal impurities inhibiting decomposition of H₂O₂.

As the reviewer suggested, we performed CH₄ oxidation experiment using an excess amount of c-H₂O₂ (10 times more than that of c-H₂O₂). Even with excess H₂O₂, no liquid products were detected after the reaction (Figure R9). This result supports that molecular H₂O₂ cannot activate CH₄ at room temperature. In addition, during revision, we performed additional experiments (see Figure R1-2) which suggest activation of CH₄ by ROSs radicals, not H₂O₂ directly. We apologize for the confusing description, and now the manuscript is revised to discuss active species more. Thank you for your comment.

Figure R9. ¹H-NMR analysis after reaction between excess amount of commercial H₂O₂ and CH₄ at room temperature. The peaks at 0.17, and 2.71 ppm are attributed to CH₄ and (CH₃)₂SO (DMSO), respectively.

Above finding is added to the manuscript (page 12 line 19-page 13 line 3 and Supplementary Fig. 10) as below. Thank you for your comment.

[page 12 line 19-page 13 line 3] *In contrast with EMPO, no liquid products were formed when c-H₂O₂ was added in the reaction cell at 25 °C (Fig. 2a and Supplementary Fig. 9-10) implying that H₂O₂ molecule itself cannot react with CH₄ and ROSs formed during electrochemical ORR are crucial for the EMPO.*

5. Where did the O atoms in the products come from, O₂ or H₂O?

Response: Thank you for your comment on the origin of O atoms. To identify the origin of O atoms in products such as CH₃OOH, CH₃OH, and HCOOH, we conducted the EMPO reaction replacing O₂ with Ar. If the O atoms come from H₂O molecules, the oxygenated products should be detected without O₂ supply because H₂O is still present in the electrolyte. However, if the O atoms in products come from O₂, we cannot detect any reaction products without O₂ supply. Figure R10 shows that no liquid products were detected after the EMPO reaction feeding Ar+CH₄ except O₂, supporting that O atoms from O₂ gas participate in activating CH₄. Furthermore, ROS experiments suggest that radical species generated from electrochemical O₂ reduction reaction activate CH₄ to form oxygenate products. Therefore, we suggest that O atoms come from O₂ not H₂O. This result matches well with the modified EMPO reaction mechanism in response #3.

Figure R10. ¹H-NMR analysis after EMPO reaction replacing O₂ with Ar. The peaks at 0.17 and 2.71 ppm are attributed to CH₄ and (CH₃)₂SO (DMSO), respectively.

Above finding is added to the manuscript (page 13 line 5-8 and Supplementary Fig. 11). Thank you for your comment.

[page 13 line 5-8] To understand the role of O₂ in the EMPO reaction, we applied the same cathodic potential in the H-cell under CH₄ and Ar flow. No CH₄ oxidation products were detected after the reaction feeding CH₄+Ar except O₂ (Supplementary Fig. 11), supporting that O atoms from O₂(g) not from the H₂O electrolyte participate in activating CH₄.

6. What was the advantages of the activated carbon electrocatalyst and how did it work?

Response: We applied an a-KB electrocatalyst in this study for the following three advantages. First, a metal-free carbon catalyst is stable under acidic conditions. The a-KB catalyst showed stable performance for EMPO reaction. Second, we focused on the investigation of the EMPO reaction mechanism. To realize this, an electrocatalyst that shows high H₂O₂ selectivity in a wide potential window is beneficial to apply. The a-KB catalyst showed high selectivity of over 90% in a wide potential range (-0.1~0.35 V vs. RHE). Third, carbon materials, especially ketjen black, are earth-abundant and cheap substances, and the synthesis process is simple, implying that it can be a universally applicable electrocatalyst. Due to the above advantages, we decided that a metal-free carbon-based catalyst, a-KB, is appropriate to investigate the EMPO system. During revision, we also test an Au foil instead of a-KB electrocatalyst, which supports that EMPO reaction is not limited to the carbon-based catalyst. It functions as an ORR electrocatalyst that converts molecular O₂ to H₂O₂ and further reduces H₂O₂ to active OH radical which can activate the methane C-H bond. Additionally, activated carbon applied cathode can reduce CH₃OOH, an unstable intermediate product, to CH₃OH enhancing the product selectivity of HCOOH. Thank you for your comment.

7. The stability test seemed for the electrocatalyst stability, out of the focus of the study? The authors need to develop a flow reactor for the EMPO approach?

Response: According to the reviewer's suggestion we developed a flow reactor for the EMPO approach to test electrocatalyst stability. As the reviewer mentioned, electrocatalysis usually shows catalytic stability by one run over long operation time. Since high concentration of HCOOH induce over-oxidation to CO₂ (Figure R3), we established the electrolyte-flowing H-cell setup to keep HCOOH concentration low as shown in Figure R11. Then, no decrease in the current density and production rate of HCOOH was observed during 6 hours of EMPO reaction time. (Figure R12). Thank you for your comment.

Figure R11. The electrolyte-flowing H-cell setup for stability operation experiment.

Figure R12. Stability test of the EMPO system. Current density, and product amount and production rate of EMPO reaction during 6 hours. Reaction conditions: 25 °C, 1 bar, O_2 : 100 sccm, CH_4 : 100 sccm, 550 mL of 0.05 M H_2SO_4 circulated by 10 mL min^{-1} , stirred at 700 rpm, and 0 V (vs. RHE).

We revised our manuscript including the details of the results and the experimental method (page 18 line 12-16, page 20 line 14-15, Fig. 5, and Supplementary Fig. 20-21). Thank you for your comment.

[page 18 line 12-16] A stability test of the EMPO system was performed (Fig. 5). The stability test was investigated under the electrolyte-flowing H-cell setup to keep HCOOH concentration low because high concentration of HCOOH induces over-oxidation to CO₂ (Supplementary Fig. 20-21). During six hours of the EMPO reaction, no decreases in the current density or production rate of HCOOH were observed demonstrating its stable operation.

[page 20 line 14-15] This EMPO system applying metal-free carbon catalyst showed significant stable performance for 6 hours.

8. In connection to Comment 7: More parameters, such as CH₄ and O₂ flow rates, and CH₄:O₂ ratios, need to be examined on the reaction? Do the solubility of CH₄ and O₂ affect the reaction?

Response: Thank you for your comment that identifying parameters that can affect the EMPO reaction. We performed additional experiments to identify the influence of flow rate and partial concentrations of gases on EMPO reaction. First, we increased the flow rate of both feeding gases from 100 sccm to 200 sccm. As you can see in Figure R13, no difference in product amount was observed meaning that total flow rates do not have an impact on the EMPO reaction. It suggested that the amount of the CH₄ and O₂ supply are sufficient and not a limiting factor.

Figure R13. The product amount of HCOOH depending on the total flow rate

On the other hand, the production of HCOOH is significantly influenced by changes in the partial concentrations of CH₄ and O₂ (Figure R14). To control the CH₄:O₂ ratio, we reduced the CH₄ flow rate to 25 sccm while maintaining the O₂ flow rate (resulting in a decrease in the

partial pressure of CH₄), which led to a decrease in the amount of HCOOH produced. In contrast, when the flow rate of O₂ was reduced to 25 sccm while keeping the CH₄ flow rate constant, the production of HCOOH increased. As the flow rate of CH₄/O₂ changed from 25 sccm/100 sccm to 50 sccm/50 sccm to 100 sccm/25 sccm, the partial concentration of CH₄ increased from 20% to 50% to 80%. The amount of HCOOH produced increased in the order of increasing partial concentration of CH₄, indicating that CH₄ was the limiting reagent and affected the reaction rate under these EMPO conditions.

Figure R14. The product amount of HCOOH depending on the flow rate ratio of CH₄:O₂.

Above findings are added to the manuscript (page 10 line 22-page 11 line 12, page 24 line 3-4, and Supplementary Fig. 8). Thank you for your comment.

[page 10 line 22-page 11 line 12] Then other parameters, which can affect production rates, such as flow rate and applied potential were examined. First, we increased the flow rate of both feeding gases (CH₄ and O₂) from 100 sccm to 200 sccm (Supplementary Fig. 8a). The same product amounts were observed meaning that these total flow rates do not have an impact on the EMPO reaction rate (i.e., sufficient supply of reactant gases). On the other hand, the production of HCOOH is significantly influenced by changes in the partial concentrations of CH₄ and O₂ (Supplementary Fig. 8b). To control the CH₄:O₂ ratio, we reduced the CH₄ flow rate to 25 sccm while maintaining the O₂ flow rate (resulting in a decrease in the partial pressure of CH₄), which led to a decrease in the amount of HCOOH produced. In contrast, when the flow rate of O₂ was reduced to 25 sccm while keeping the CH₄ flow rate constant, the production of HCOOH increased. As the flow rate of CH₄/O₂ changed from 25 sccm/100 sccm

to 50 sccm/50 sccm to 100 sccm/25 sccm, the partial concentration of CH₄ increased from 20% to 50% to 80%. The amount of HCOOH produced increased in the order of increasing partial concentration of CH₄, indicating that CH₄ was the limiting reagent and affected the reaction rate under these EMPO conditions.

[page 24 line 3-4] For the flow rate experiments, the flow rates of CH₄ and O₂ were varied while all other reaction conditions were maintained.

Reviewer #4: In the submitted manuscript, the authors developed an electrochemical system to valorize methane into formate as a liquid value-added chemical. They carried out methane conversion by first reducing oxygen to peroxide via a carbon catalyst, which then reacted with CH₄ in a chemical step to generate CH₃OOH. This species was then reduced to CH₃OH electrochemically and oxidized chemically again by the peroxide to yield HCOOH as the final product.

The system is rather simple and proposed mechanism quite reasonable and supported with an intuitive set of experiments. This is a significant value-added as electrochemical methane oxidation systems are not well established and the few that do exist often require harsh conditions and feature low yields. Because of this, I would recommend publication in Nature Communications with several suggestions below.

1. Introduction – would be worth mentioning the carbon footprint of steam reforming vs electrochemical valorization of methane.

Response: Thank you for your comment that mentioning carbon footprint of steam reforming vs electrochemical valorization of methane would be worth. We added that steam methane reforming (SMR) has high emission of CO₂ and methane partial oxidation could mitigate CO₂ emission in the introduction (*Environ. Sci. Technol.* **53**, 7103-7113 (2019)) (page 4 line 8-10, page 5 line 3-6, and page 6 line 4-5).

[page 4 line 8-10] The methane partial oxidation to liquid oxygenates such as CH₃OH and HCOOH has been regarded as a promising approach for effective natural gas utilization as it can facilitate transportation and storage and mitigate CO₂ emission.

[page 5 line 3-6] Since SMR process has high emission of CO₂, at almost 9 kg of CO₂ per 1 kg of H₂ produced, and requires energy-intensive and large infrastructure, it is infeasible at remote geographical locations.^{10, 11} Therefore, energy-efficient, low CO₂ emission, and miniaturized technologies are highly required for direct partial oxidation of methane.

[page 6 line 4-5] It offers a sustainable route for converting CH₄ to liquid fuels, mitigating CO₂ emissions.

2. Why is such a large overpotential required for ORR? State-of-the-art carbon-based catalysts

can catalyze this reaction near the thermodynamic onset potential of approx. 0.7V vs. RHE.

Response: We agree with your opinion that it is worth applying state-of-the-art carbon-based catalysts to improve EMPO reaction efficiency in terms of the reaction potential. However, we applied an a-KB electrocatalyst in this study for the following two purposes. First, we focused on the investigation of the EMPO reaction mechanism, not on the electrocatalyst. To realize this, an electrocatalyst that shows high H₂O₂ selectivity in a wide potential window is needed. The a-KB catalyst showed high selectivity of over 90% in a wide potential range (-0.1~0.35 V vs. RHE). Second, a metal-free carbon catalyst is stable under the acidic condition. If we applied metal-nitrogen-doped carbon (MNC) catalyst such as CoNC, we cannot guarantee the stability of the EMPO system under acidic reaction conditions even if they can show higher EMPO reaction efficiency in the short term. The a-KB catalyst showed stable performance for six hours of EMPO reaction. It is a good suggestion to apply state-of-the-art carbon-based catalysts such as N-doped carbon, and oxidized carbon nanotube to improve the production efficiency of the EMPO system. Thank you for your comment.

3. The collection efficiency of 37% (Page 19, line 331) is higher than a typical collection efficiency of 25%, based on the reviewer's experience. Please provide a technical and commercial details of the RRDE electrode. If it is possible, please provide a collection efficiency measurement in the supplementary information.

Response: Thank you for your suggestion to confirm the collection efficiency. Technical and commercial details of the RRDE electrode that we used in this study are given below:

E7R9 RRDE (Pine research)

Ring-Disk Gap = 320 μm; Collection Efficiency = 37%

Disk OD = 5.61 mm; Ring OD = 7.92 mm; Ring ID = 6.25 mm

Fixed-Disk RRDE tip; 15.0 mm OD PTFE shroud

Additionally, we measured the collection efficiency (N) of our RRDE by using the [Fe(CN)₆]^{3-/4-} redox system. The catalyst-deposited RRDE was soaked in N₂-saturated 0.1 M KOH + 2 mM K₃[Fe(CN)₆], and chronoamperometry was performed at -0.3 V (vs. Ag/AgCl) while the ring potential was fixed at 0.5 V (vs. Ag/AgCl) for 60 s (Figure R1). The background current was also obtained similarly, but the applied disk potential was 0.5 V (vs. Ag/AgCl). The

collection efficiency could be calculated as follows:

$$N = \frac{|i_r - i_{r,bg}|}{i_d}$$

Figure R1. Background-corrected chronoamperometric response of the RRDE at -0.3 V (vs. Ag/AgCl) with an electrode rotation speed of 1600 rpm measured in 2 mM $K_3[Fe(CN)_6]$ + 0.1 M KOH electrolyte.

Where i_r , $i_{r,bg}$, and i_d denote the ring current, background ring current, and disk current, respectively. The result yields that the collection efficiency is 37%, which is similar to the value provided by the manufacturer.

Thank you for your comment and the results and details of the experimental method are included in the method section of the main manuscript (page 21 line 18-21, page 22 line 17- page23 line 3, and Supplementary Fig. 23).

[page 21 line 18-21] For the rotating ring disk electrode (RRDE) measurements, a three-electrode system was constructed with an RRDE (Pine research, E7R9 RRDE) (glassy carbon (GC) disk + Pt ring), a Ag/AgCl (stored in 3 M KCl) reference electrode, and a Pt foil counter electrode.

[page 22 line 17-page 23 line 3] The collection efficiency (N) was determined using the $[Fe(CN)_6]^{3-/4-}$ redox system. Chronoamperometry was carried out at -0.3 V (vs. Ag/AgCl) while the ring potential was fixed at 0.5 V (vs. Ag/AgCl) for 60 s on the catalyst-deposited RRDE in N_2 -saturated 0.1 M KOH (Sigma-Aldrich, 90%) + 2mM $K_3[Fe(CN)_6]$ (Sigma-Aldrich, $\geq 99.0\%$). The background current was obtained similarly, but the disk potential was 0.5 V (vs. Ag/AgCl).

The collection efficiency was calculated as follows:

$$N = \frac{|i_r - i_{r,bg}|}{i_d}$$

Where i_r , $i_{r,bg}$, and i_d denote the ring current, background ring current, and disk current, respectively. The collection efficiency was 37% (Supplementary Fig. 23).

4. It seems the authors used molar fraction selectivity and faradaic efficiency interchangeably. The H_2O_2 selectivity formula in page 19 is based on the molar fraction selectivity, which is always higher than faradaic efficiency. (Nature Catalysis, 2020, 3, 605-607). The reported selectivity of 90% in Supplementary Figure 3, is the same with FE mentioned in Page 21 line 367. Please keep consistent

Response: We corrected our misrepresentation according to your recommended article (*Nat. Catal.* **3**, 605-607 (2020)) carefully. The revised values can be seen in below (page 22 line 12, page 22 line 15-16, and page 25 line 9). Thank you for your comment.

[page 22 line 12] H_2O_2 Selectivity (%) = $200 \times \frac{i_r/N}{i_d+i_r/N}$

[page 22 line 15-16] The Faradaic efficiency of H_2O_2 was calculated by equation below:

$$F.E._{H_2O_2} (\%) = 100 \times \frac{i_r/N}{i_d}$$

[page 25 line 6-10] The amount of H_2O_2 supplied was determined based on RRDE analysis.

$$\text{The amount of } H_2O_2 = \frac{F.E._{H_2O_2} \times j \times A \times t}{n \times F}$$

where, $F.E._{H_2O_2}=81.8\%$, j is current density, A is area of electrode, t is reaction time, n is moles of electron to produce 1 mole of H_2O_2 , and F is faraday constant.

5. Page 13 line 214 mentions “These results suggest that CH_3OH and $HCOOH$ cannot be reduced on the a-KB cathode under these conditions, and the current increase observed in Fig. 2b is due to the reduction of CH_3OOH .” On the other hand, figure 2f shows that the increasing $HCOOH$ concentration enhances the reduction current, though to a smaller extent. What is

occurring here?

Response: The reviewer asked about the reduction current density trend of the oxygenated products. In comparison with CH₃OOH, only capacitive currents were observed when various concentrations of CH₃OH and HCOOH were applied. The capacitive current is affected by double layer thickness. With increasing double layer thickness, the current density is decreased. The HCOOH may affect double layer thickness inducing decrease capacitive current upon increasing HCOOH concentration. Thank you for your comment.

We added this discussion in supplementing information (Supplementary Fig. 14).

[Supplementary Figure 14] In comparison with CH₃OOH, only capacitive currents were observed when various concentrations of CH₃OH and HCOOH were applied. The capacitive current is affected by double layer thickness. With increasing double layer thickness, the current density is decreased. The HCOOH may affect double layer thickness inducing decrease capacitive current upon increasing HCOOH concentration.

6. It seems the data in Figure 2d, 2e, and 2f were recorded by RRDE. If so, please provide the ring current on the figures in supplementary information.

Response: Thank you for your comment. The detailed experiment conditions are now included in the revision manuscript. The reducibility tests of CH₃OOH, CH₃OH, and HCOOH were performed using a working electrode prepared by spraying a-KB based catalyst ink on carbon paper (11.25 cm²). We did not use the RRDE. We measured reduction currents of the working electrode with various concentrations (0-40 mM) of reactants.

[page 27 line 9-10] A working electrode prepared by spraying a-KB based catalyst ink on carbon paper (11.25 cm²) was used.

7. Please have a look at “ACS Catalysis 2018, 8, 9, 7961-7972”, arguing peroxide assisted methane oxidation via a Fenton reaction. In there, CH₃OH formation is considered to happen in parallel to CH₃OOH production, with multiple possible paths. No formic acid was observed. It is better to explain the difference between that article and this study.

Response: Thank you for your comments on the reaction pathways. The reference paper (ACS Catalysis 2018, 8, 9, 7961-7972) studied the mechanism of methane oxidation via a Fenton reaction based on the density functional theory (DFT) simulation. Among possible reaction mechanisms, Fenton-type reaction involves free $\cdot\text{OH}$ radicals from H_2O_2 as the catalytic species which can activate CH_4 to CH_3OH or CH_3OOH . Although the reaction step simulation to HCOOH formation was not included in the reference paper, many researchers experimentally reported that the reactive $\cdot\text{OH}$ radicals further oxidized CH_3OH to HCOOH (*Angew. Chem. Int. Ed.* **51**, 5129-5133 (2012), *Angew. Chem. Int. Ed.* **52**, 1280-1284 (2013), *Angew. Chem. Int. Ed.* **55**, 13441-13445 (2016), *ACS Catal.* **11**, 6684-6691 (2021)).

During the revision, we also investigated the reaction pathway whether reactive oxygen species (ROSs) were involved in the EMPO process. As discussed below, we propose that the electrochemical ORR process generates ROSs which play as active species for CH_4 oxidation, similar to the reference reports. Meanwhile, we demonstrated that CH_3OH can be also oxidized to HCOOH by *in-situ* generated H_2O_2 in our study. Based on the below experimental results, we modified the main manuscript and proposed mechanism pathways.

Since $\cdot\text{OH}$ radicals can be formed during electrochemical oxygen reduction reaction (ORR) (*Angew. Chem. Int. Ed.* **60**, 10375-10383 (2021) / *J. Phys. Chem. Lett.* **12**, 7797-7803 (2021)), we conducted electron paramagnetic resonance (EPR) analysis by adding 5,5-dimethyl-1-pyrroline N-oxide (DMPO) in the electrolytes as a spin trap to verify radical species directly. EPR signals were collected by measuring electrolytes before and after applying reduction potential (0 V vs. RHE) flowing O_2 and O_2+CH_4 , respectively, with adding DMPO (Figure R2). While no radical was observed in the electrolyte before the reaction (grey line), a quartet spectrum with a relative intensity ratio of 1:2:2:1 appeared after applying potential under O_2+CH_4 flowing (blue line). It is typically attributed to the formation of DMPO-OH adducts (*Free Radic. Biol. Med.* 1987, 3, 259-303) confirming that radicals were formed during EMPO reaction. Especially, DMPO-OH adducts were also detected when only O_2 was fed except CH_4 showing that ROSs are generated during electrochemical ORR reaction (red line).

Figure R2. EPR spectra of the electrolytes before and after applying potential under O₂ and O₂+CH₄ flowing with DMPO.

Since we confirmed that ·OH radicals are produced during the EMPO reaction, we agreed with your comment that CH₃OH formation happen in parallel with CH₃OOH generation and revised our proposed reaction mechanism as below (Figure R3):

ROSs formation:

CH₄ activation:

CH₃OOH formation:

CH₃OH formation:

Figure R3. Proposed mechanism. Schematic illustration of the reaction mechanism for selective HCOOH production by EMPO.

We revised the manuscript and the proposed mechanism based on the above results (page 3 line 13-15, page 7 line 1-2, page 12 line 19-page 13 line 5, page 13 line 9-19, page 16 line 5-13, page 20 line 7-8, page 20 line 10-12, page 24 line 7-10 page 26 line 15-19, and Fig 2a, 3). Thank you for your comment.

[page 3 line 13-15] The mechanistic study revealed that reactive oxygen species (ROSs) such as $\cdot OH$ and $\cdot OOH$ radicals are produced during the reaction and activated methane and methanol.

[page 7 line 1-2] Reactive oxygen species (ROSs) (i.e., $\cdot OH$ and $\cdot OOH$ radicals) were generated during ORR and activated CH_4 and CH_3OH .

[page 12 line 19-page 13 line 5] In contrast with EMPO, no liquid products were formed when $c\text{-}H_2O_2$ was added in the reaction cell at 25 °C (Fig. 2a and Supplementary Fig. 9-10) implying that H_2O_2 molecule itself cannot react with CH_4 and ROSs formed during electrochemical ORR are crucial for the EMPO.^{37, 38}

We performed control experiments to figure out whether ROSs are generated from O_2 and participate in the EMPO reaction.

[page 13 line 9-19] Then, to verify radical species more directly, we additionally conducted electron paramagnetic resonance (EPR) analysis by adding 5,5-dimethyl-1-pyrroline N-oxide

(DMPO) in the electrolytes as a spin trap. EPR signals were collected by measuring electrolytes before and after applying reduction potential (0 V vs. RHE) flowing $O_2(g)$ or $O_2(g)+CH_4(g)$ under the addition of DMPO (Fig. 2b). We confirmed that no radical signal was observed in the electrolyte before the reaction (grey line). Meanwhile, a quartet spectrum with a relative intensity ratio of 1:2:2:1 appeared after applying potential under O_2+CH_4 feeding (blue line). It is typically attributed to the formation of DMPO–OH adducts supporting that radicals were formed during EMPO reaction.³⁹ Especially, DMPO–OH adducts were also detected when only O_2 was fed without CH_4 suggesting that ROSs are generated as the result of electrochemical ORR in our system (red line).

[page 16 line 5-13] As illustrated in Fig. 3, first, O_2 is reduced to H_2O_2 on the electrode through the electrochemical two-electron pathway ORR, and ROSs are formed (ROSs formation). Second, $\cdot OH$ radicals activate CH_4 to produce $CH_3\cdot$ radicals in the electrolyte (CH_4 activation). Then, CH_3OOH is generated by the reaction between $CH_3\cdot$ and $\cdot OOH$ radicals (CH_3OOH formation). Subsequently, CH_3OH was generated by electrochemical reduction of CH_3OOH and radical reaction between $CH_3\cdot$ and $\cdot OH$. Finally, $HCOOH$ was formed in the presence of $\cdot OH$ radicals ($HCOOH$ formation). Thus, $HCOOH$ can be generated as a selective reaction product with the aid of electrochemical reduction potential at room temperature and atmospheric pressure in the EMPO system.

[page 20 line 7-8] CH_4 can be activated by the in-situ cathodically generated ROSs (i.e., $\cdot OH$ and $\cdot OOH$ radicals) from the O_2 reduction.

[page 20 line 10-12] Through a mechanistic study of the EMPO process, we propose that ROSs are the active species of the partial oxidation of CH_4 .

[page 24 line 7-10] For the EPR experiment, 1mmol of DMPO (Sigma-Aldrich) was applied as a spin trap before the reaction. Other reaction conditions were the same except for feeding O_2 or O_2+CH_4 gases.

[page 26, line 15-19] **EPR experiment**

The EPR spectrum of electrolytes was measured at KBSI Seoul Western Center using CW/Pulse EPR system with the following parameters: frequency 9.852 GHz; power 3 mW; modulation amplitude 1 G; time constant 20.48 ms; conversion time 20.00 ms; scan 8; temperature RT.

REVIEWER COMMENTS

Reviewer #2 (Remarks to the Author):

The authors have addressed the issues from reviewers, this manuscript can be accepted.

Reviewer #3 (Remarks to the Author):

In the revised submission the authors replied appropriately to my previous comments and revised the manuscript accordingly. I recommend to accept the revised manuscript.

Comments from Reviewer #3 on Behalf of Reviewer #1:

Most of the author's replies to Reviewer #1 are satisfying. However, the authors failed to reply appropriately to part of the comment 1- "The authors attributed this to the presence of stabilizers that inhibit the decomposition of H₂O₂. But, if the reaction takes place chemically between CH₄ and H₂O₂, then why do we need H₂O₂ to decompose? Heating the system to 70 degrees forces H₂O₂ to decompose and form ROSs which will attack CH₄ to form CH₃OOH. A report by Jin et al. (Science 367, 193-197 (2020)) showed much better performance at reaction temperatures at low as 70."

The authors are suggested to run the reaction with H₂O₂ thermally at elevated temperatures, whose results can be compared to their results of electro-assisted methane oxidation via in-situ cathodically generated H₂O₂ under ambient conditions. In my opinion, with such information, the authors will reply adequately to all comments of Reviewer #1.

Reviewer #4 (Remarks to the Author):

The authors have performed a set of supplementary experiments that helped elucidate the mechanism of partial oxidation. I believe that this addition strengthens the manuscript significantly and thus I would recommend its acceptance.

Responses to Reviewers' comments

We appreciate the reviewers for the valuable comments which improve our manuscript. We revised the manuscript thoroughly based on the comments, and the changes in the revised manuscript are highlighted in red.

Reviewer(s)' Comments to Author:

Reviewer #2: The authors have addressed the issues from the reviewers, this manuscript can be accepted.

Reviewer #3: In the revised submission the authors replied appropriately to my previous comments and revised the manuscript accordingly. I recommend to accept the revised manuscript.

Comments from Reviewer #3 on Behalf of Reviewer #1:

Most of the author's replies to Reviewer #1 are satisfying. However, the authors failed to reply appropriately to part of the comment 1 - "The authors attributed this to the presence of stabilizers that inhibit the decomposition of H_2O_2 . But, if the reaction takes place chemically between CH_4 and H_2O_2 , then why do we need H_2O_2 to decompose? Heating the system to 70 degrees forces H_2O_2 to decompose and form ROSs which will attack CH_4 to form CH_3OOH . A report by Jin et al. (Science 367, 193-197 (2020)) showed much better performance at reaction temperatures at low as 70."

The authors are suggested to run the reaction with H_2O_2 thermally at elevated temperatures, whose results can be compared to their results of electro-assisted methane oxidation via in-situ cathodically generated H_2O_2 under ambient conditions. In my opinion, with such information, the authors will reply adequately to all comments of Reviewer #1.

Response: Thank you for your comments. As the reviewer pointed out, to answer Reviewer #1's comment 1 clearly, we reconsider two points more based on the temperature experiments

and the additional control experiments regarding ROSs; First, can molecular H₂O₂ activate CH₄ directly? According to Fig. 2a of the main manuscript and Supplementary Fig. S9-10, no liquid products were detected after the CH₄ oxidation reaction with molecular H₂O₂ (c-H₂O₂) directly at room temperature (denoted as c₂₅-H₂O₂ in Fig. 2a). In addition, trapping experiments and electron paramagnetic resonance (EPR) analysis suggested that CH₄ is activated by ROSs. Therefore, we agree that CH₃OOH production from CH₄ is achieved not by molecular H₂O₂ directly, but by ROSs in EMPO reaction. Based on these considerations, we proposed the revised reaction mechanism via ROSs.

Next, we conducted a CH₄ oxidation reaction with H₂O₂ thermally at elevated temperatures (70 °C, denoted as c₇₀-H₂O₂ in Fig. 2a) and compared it to the result of EMPO reaction at room temperature. At 70 °C, only a small amount (0.8 μmol h⁻¹) of CH₃OOH was produced implying ROSs can be formed by thermal decomposition of H₂O₂ at elevated temperatures and activate CH₄ slightly. However, its conversion was much lower compared to the EMPO system although EMPO was performed at room temperature. Furthermore, we conducted the CH₄ oxidation reaction with molecular H₂O₂ at other elevated temperatures such as 50 °C, and 90 °C, and the elevated temperature conditions similarly showed that only CH₃OOH was produced (Figure R1). Increasing reaction temperatures from room temperature to 50 and 70 °C enhances the production rates (Figure R2) because the thermal formation of ROSs can activate CH₄ to form CH₃COOH. Meanwhile, the production rate of the unstable CH₃OOH rather decreases as the reaction temperature increases from 70 °C to 90 °C. To sum up, molecular H₂O₂ can be decomposed to ROSs at elevated temperatures. However, CH₃OOH is the only product, and the production rates of CH₃OOH at elevated temperatures are much lower without the electrocatalytic reaction, demonstrating the EMPO system is more effective to produce liquid products (CH₃OOH, and HCOOH) at room temperature. We hope our answer can now adequately resolve all previous comments from the reviewers.

Figure R1. ¹H-NMR analyses after CH₄ oxidation reaction with commercial H₂O₂ at elevated temperatures (50-90 °C). The peaks at 0.17, 2.71, and 3.85 ppm are attributed to CH₄, (CH₃)₂SO (DMSO), and CH₃OOH, respectively.

Figure R2. CH₃OOH production rates after CH₄ oxidation reaction with commercial H₂O₂ at elevated temperatures (50-90 °C).

Above findings are added to the manuscript (page 14 line 9-12, Supplementary Fig. 13-14). Thank you for your comments.

[page 14 line 6-13] Then the reaction temperature was increased to 70 °C to activate *c*-H₂O₂. At 70 °C, the *c*-H₂O₂ reacted with CH₄, but only unstable CH₃OOH was produced at 0.8 μmol h⁻¹ production rate (Supplementary Fig. 13), which was much lower production than that of the EMPO system at room temperature (Fig. 2a). *c*-H₂O₂ reacted with CH₄ were also tested at other reaction temperatures (50-90 °C), and all showed only a small amount of CH₃OOH production (Supplementary Fig. 14). The production rate of the unstable CH₃OOH rather decreased as the reaction temperature increased from 70 °C to 90 °C. This supports that the electrocatalytic condition (*e*-H₂O₂) contributes to the increase in liquid products.

Reviewer #4: The authors have performed a set of supplementary experiments that helped elucidate the mechanism of partial oxidation. I believe that this addition strengthens the manuscript significantly and thus I would recommend its acceptance.

REVIEWERS' COMMENTS

Reviewer #3 (Remarks to the Author):

The authors added experimental results and discussion to clarify the reviewer's comments in the revised manuscript. The revised manuscript can be accepted.